# A physicochemical perspective of aging from single-cell analysis of pH, macromolecular and organellar crowding in yeast

Sara N Mouton[1], David J Thaller[2], Matthew M Crane[3], Irina L Rempel[1], Owen T Terpstra[1], Anton Steen[1], Matt Kaeberlein[3], C Patrick Lusk[2], Arnold J Boersma[4]*, Liesbeth M Veenhoff[1]*

[1]European Research Institute for the Biology of Ageing, University of Groningen, University Medical Center Groningen, Groningen, Netherlands; [2]Department of Cell Biology, Yale School of Medicine, New Haven, United States; [3]Department of Pathology, School of Medicine, University of Washington, Seattle, United States; [4]DWI-Leibniz Institute for Interactive Materials, Aachen, Germany

**Abstract** Cellular aging is a multifactorial process that is characterized by a decline in homeostatic capacity, best described at the molecular level. Physicochemical properties such as pH and macromolecular crowding are essential to all molecular processes in cells and require maintenance. Whether a drift in physicochemical properties contributes to the overall decline of homeostasis in aging is not known. Here, we show that the cytosol of yeast cells acidifies modestly in early aging and sharply after senescence. Using a macromolecular crowding sensor optimized for long-term FRET measurements, we show that crowding is rather stable and that the stability of crowding is a stronger predictor for lifespan than the absolute crowding levels. Additionally, in aged cells, we observe drastic changes in organellar volume, leading to crowding on the micrometer scale, which we term organellar crowding. Our measurements provide an initial framework of physicochemical parameters of replicatively aged yeast cells.

*For correspondence: boersma@dwi.rwth-aachen.de (AJB); l.m.veenhoff@rug.nl (LMV)

## Introduction

Cellular aging is a process of progressive decline in homeostatic capacity (*Gems and Partridge, 2013*; *Kirkwood, 2005*). Key molecules have been identified to govern the aging process and can extend health and lifespan by maintaining prolonged homeostasis (*Kenyon, 2010*). The function of these molecules ultimately depends on physicochemical properties, such as pH, macromolecular crowding, and ionic strength. As all these physicochemical properties require maintenance, the question how stable these properties are in aging is pertinent. Identifying age-related changes in the cytosol, where all proteins are synthesized and most metabolic processes take place is crucial to understanding how aged cells function differently from their younger counterparts. *Saccharomyces cerevisiae* is an excellent model system to quantify physicochemical changes during aging, as single cells can be directly monitored by microscopy as they age (*Crane et al., 2014*; *Jo et al., 2015*). Importantly, many of the molecular mechanisms that contribute to yeast aging are conserved in humans (*Janssens and Veenhoff, 2016a*).

pH homeostasis is an important parameter in human aging, as human senescent cells show increased lysosomal pH (*Kurz et al., 2000*), and in age-related pathologies such as Alzheimer's and Parkinson's disease, lysosomes are dysfunctional (*Carmona-Gutierrez et al., 2016*). The main proton pumps in the lysosomal membrane (termed vacuole in yeast), the V-ATPases, are highly conserved

from yeast to human, and Pma1 - the yeast plasma membrane ATPase, shares structural and functional similarities with the Na⁺K⁺ ATPases in mammalian cells (*Forgac, 2007*; *Morth et al., 2011*; *Nelson et al., 2000*). Pma1 localizes in the plasma membrane and transports cytosolic protons out of the cell (*Ferreira et al., 2001*; *Orij et al., 2011*; *Serrano et al., 1986*), while the V-ATPase pumps protons from the cytosol into the lumen of various organelles and regulates their pH (*Forgac, 2007*; *Kane, 2006*). Both enzymes change in aging: Pma1 levels increase as this protein is asymmetrically retained in the mother cell (*Henderson et al., 2014*) and the components of the V-ATPase become substoichiometric (*Janssens et al., 2015*), potentially reducing the number of functional complexes. Concomitantly, changes in vacuolar and cytosolic pH have been reported in aging, namely, an alkalinization of the cortex (region close to the plasma membrane) (*Henderson et al., 2014*), and alkalinization of the vacuole (*Chen et al., 2020*; *Hughes and Gottschling, 2012*), both measured in single cells and occurring early in the lifespan. In addition, in a population-based study, an acidification of the cytosol at the end of the replicative lifespan was reported (*Knieß and Mayer, 2016*). So, while there is evidence for changes in pH in cellular aging, what is currently missing is a single-cell perspective on cytosolic pH in yeast replicative ageing.

Human senescent cells and aged yeast cells increase in size, which might result in dilution of the cytoplasm and changes in macromolecular crowding (*Neurohr et al., 2019*). Cells are highly crowded, with macromolecular concentrations estimated to be between 80 and 400 mg/mL (*Cayley et al., 1991*; *Zimmerman and Trach, 1991*). Macromolecular crowding retards diffusion, influences protein volume and association equilibria (*Dix and Verkman, 2008*; *Ellis, 2001*; *Zhou et al., 2008*), including, for example condensate formation in vitro and in vivo (*Delarue et al., 2018*; *Woodruff et al., 2017*). These effects are caused by steric exclusion, next to weak chemical interactions (*Gnutt and Ebbinghaus, 2016*; *Rivas and Minton, 2016*; *Sarkar et al., 2013*), and depend on the concentration, size, and shape of the molecules involved and are larger when crowders are smaller sized than the reacting molecule (*Marenduzzo et al., 2006*; *Rivas and Minton, 2016*). For example, an increased number of ribosomes slows down diffusion of 20 nm and 40 nm particles, but not average-sized proteins (*Delarue et al., 2018*). The propensity to undergo a phase transition to a 'gel-like' state is amplified by macromolecular crowding (*Joyner et al., 2016*) and also influenced by the pH: Starvation of yeast cells leads to acidification of the cytoplasm, which leads to a phase transition that hampers diffusion of μm-sized particles (*Munder et al., 2016*). Therefore, quantification of macromolecular crowding on the single-cell level could provide evidence on whether crowding could be a driver of aberrant biochemical organization in aging.

The volume of a cell needs to be coupled to biopolymer synthesis in order to maintain macromolecular crowding (*Burg, 2000*; *Minton et al., 1992*; *van den Berg et al., 2017*; *Zimmerman and Minton, 1993*). For example, it has been suggested that oversized yeast cells have reduced molecular density (*Neurohr et al., 2019*). One of the most dramatic features of aged yeast cells is a marked increase in cell size (*Fehrmann et al., 2013*; *Janssens et al., 2015*; *Lee et al., 2012*). Concomitantly, yeast organelles like vacuoles (*Lee et al., 2012*), the nucleus (including nucleoli *Crane et al., 2019*; *Kennedy et al., 1997*), and mitochondria (*Scheckhuber et al., 2007*) can exhibit changes to their respective morphology. These changes in compartment size and shape directly impact the cytosolic volume and additionally present physical barriers to molecular movement and surfaces on which molecules can be adsorbed. Furthermore, changes in compartmental volume alters energy consumption. For example, a small compartment, such as the endosome, needs to import fewer protons to maintain pH compared to larger compartments (*Luby-Phelps, 1999*). Additionally, specific cell types have different organelle sizes related to their function: for example, secretory cells have expanded ER (*Federovitch et al., 2005*). However, despite its importance, it is not clear how much compartmental volumes change during aging.

Here, we present a framework describing how the critical and interconnected physicochemical parameters of pH and crowding change during yeast replicative aging. We find that the cytosol shows modest acidification in early aging and drops significantly after the cells stop dividing. We optimize our macromolecular crowding sensors for long-term FRET measurements and show that longer-lived cells tend to maintain macromolecular crowding better than shorter-lived cells. While macromolecular crowding changes only modestly in aging, we observe drastic changes in organellar volumes, leading to crowding on the μm scale, which we term organellar crowding. In light of our evidence, pH and crowding must be taken into account when investigating and interpreting the hallmarks of aging.

## Results

### Yeast replicative aging leads to acidification of the cytosol, especially after entry into senescence

To follow cytosolic pH levels in aging, we utilized the fluorescence-based, genetically encoded pH sensor, ratiometric pHluorin (*Miesenböck et al., 1998*). pHluorin is a GFP variant that responds to changing pH: With increasing acidity, the excitation at 390 nm decreases, while the excitation at 475 nm increases. We expressed pHluorin in the BY4741 background strain and recorded the $F_{390}/F_{475}$ ratios from aging cells using the ALCATRAS microfluidic device (*Crane et al., 2014*; *Figure 1A*). We followed single-cell life histories from 80 cells over 80 hr, taking widefield fluorescent images every 10 hr and bright-field images every 20 min. Additionally, we performed an in vivo calibration of pHluorin with cells loaded in the microfluidic device (*Figure 1B*).

To compare differences in pH between populations of young and aged cells, we grouped the first and the last measurements taken for each cell under two categories: 'young' and 'old'. The 'young' category contains pH measurements in the first fluorescent image when cells are predominantly age 0, since newborn daughters are the most common group in an exponentially growing culture. These young cells have an average cytosolic pH of 7.7 and pH levels are comparable between individual cells (*Figure 1C*). Furthermore, the measured average cytosolic pH is higher than the previously reported pH of 7.3 (*Orij et al., 2009*), but similar to the pH reported from population-based measurements (*Knieß and Mayer, 2016*). The 'old' category contains the last pH measurement of a cell, taken less than 10 hr before its death. This category is composed of senescent and non-senescent cells of different replicative ages, where most cells had no divisions left to complete (n = 59, 100% of RLS), and others had up to five divisions to complete (n = 21, 78-96% of RLS). The average lifespan of this cohort is 23 divisions. We observe that the pH of individual old cells spans a wide range of pH values from 7.8 to 5.7, and 36% of the cells maintain a pH above 7.5, which is the lower boundary in young cells. Comparing the old and young cells, our results show that the average cytosolic pH significantly decreases by 0.5 pH units in old cells, compared to young (*Figure 1C*), corresponding with previous findings (*Knieß and Mayer, 2016*).

As our dataset is composed of single-cell life trajectories, we can analyze the intrinsic variability in cytosolic pH values within the population, the timing of changes in pH, and the correlation of these phenotypes to cell lifespan. As depicted in *Figure 1D*, we plot the pH of single-cell trajectories for different ranges of replicative lifespans. We observed a gradual decrease in pH already early in life in almost all cells, and interestingly this gradual decrease is followed by a substantial drop in pH in a subpopulation of cells that stop dividing and enter senescence (*Figure 1D*). The decrease in pH in the earlier divisions (through a replicative age of 15 ± 2) correlates to the cell lifespan ($R^2$ = 0.19, p=0.0007185) (*Figure 1—figure supplement 1C*), but we find no relationship between the lifespan of cells and pH at a young age, or the pH at old age (*Figure 1—figure supplement 1D–F*). Life trajectories of the cells with a very low pH at young or old age (the population outliers of *Figure 1C*) are excluded from 1*D* (*Figure 1—figure supplement 1A and B*). While the outliers from the aged population have nothing in common, the seven young cells, which had a lower pH than 91% of the young cohort, all increase their cytosolic pH within the next 10 hr to levels above 7.5 (*Figure 1—figure supplement 1B*). We conclude that, apart from this subpopulation of young cells with low starting pH, a shared phenotype of all aging yeast cells is that the cytosolic pH drops gradually and modestly throughout the mitotic lifespan, and that when cells stop dividing but remain alive, the pH decreases steeply.

To investigate the precise timing of entry into senescence in relation to the sudden pH decrease in the cytosol, we generated a data set with a higher time resolution where fluorescent measurements were taken in intervals of 1 hr for the total duration of 50 hr. We collected and analyzed single-cell trajectories from 50 cells (see Materials and methods section). These cells have a shorter lifespan compared to the cells from the low time resolution (*Supplementary file 3*), possibly due to higher phototoxicity. We qualitatively determine three types of behavior. Firstly, we find that 36% of the population is comprised of cells that display an abrupt entry into senescence. These cells have the highest increase in cytosolic acidity (~2-fold increase, SD = 0.4), and their entry into senescence always precedes the sharp increase in acidity (*Figure 1E*, *Figure 1—figure supplement 2A*; *Figure 1—figure supplement 3*). Hence, cytosol acidification cannot be the cause of entry into senescence in these cells. Secondly, we observe that 50% of the cells have a gradual entry into

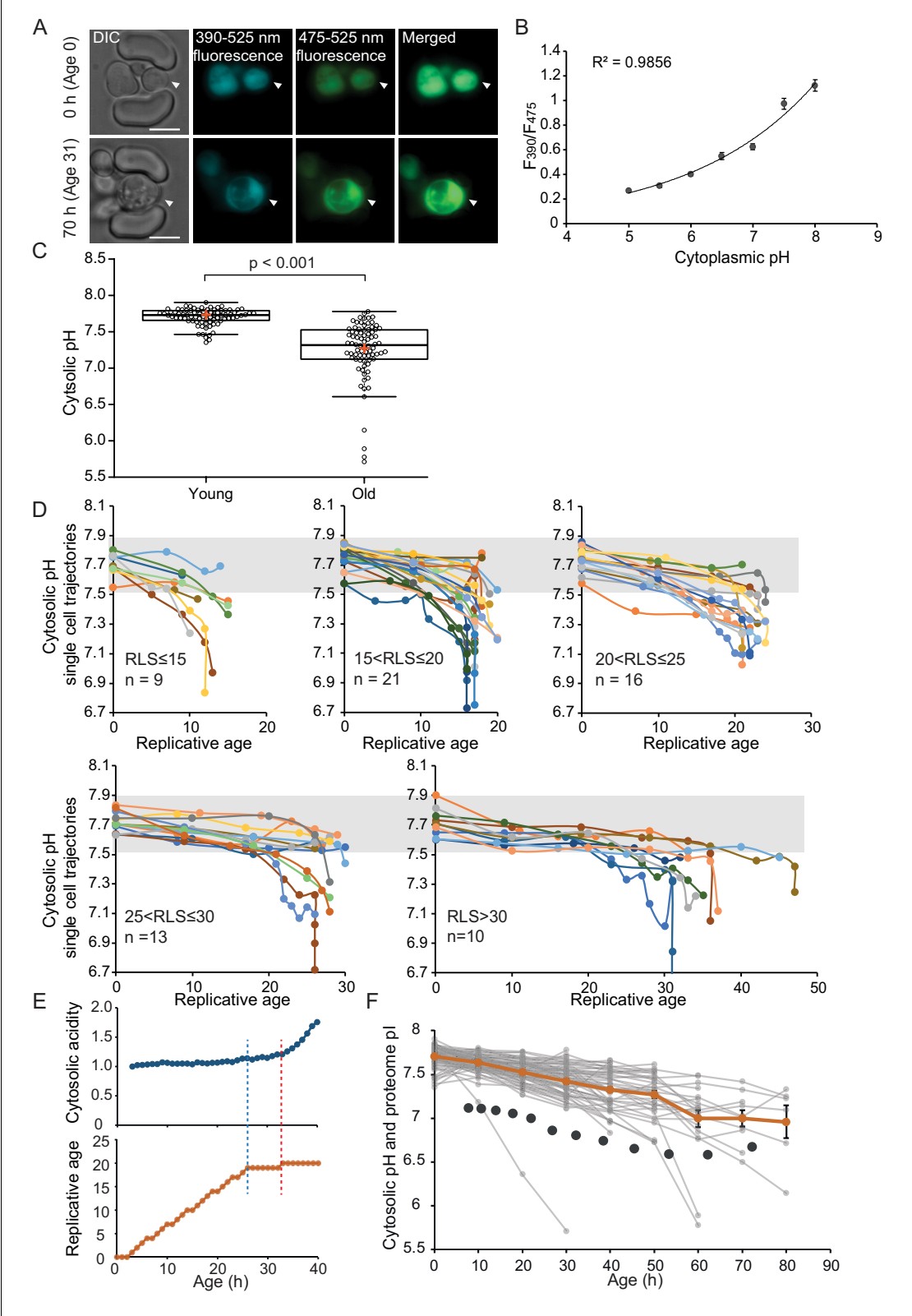

**Figure 1.** The cytosol of old cells acidifies, and cells display more substantial variability in cytosolic pH with aging, especially after entry into senescence. (**A**) Representative images of the same cell expressing ratiometric pHluorin imaged at the start of the experiment and after 80 hr; the replicative age is indicated. Young cells are trapped in the microfluidic device, and bright-field images are taken every 20 min to determine the age of the cells. Fluorescent images are taken once every 10 hr (panels, **C, D, F**) or every hour (panel **E**) with excitation at 390 and 475 nm and emission at 525

*Figure 1 continued on next page*

*Figure 1 continued*

nm. The images show DIC, DAPI, and FITC channels and merged fluorescence. The scale bar represents 5 µm. The trapped aging cell is indicated with white arrowheads. (B) Calibration curve showing the relationship between intracellular pH and pHluorin ratios. Cells expressing pHluorin from an exponential culture at $OD_{600}$ of 0.5 were resuspended in buffers, titrated to pH 5, 5.5, 6, 6.5, 7, 7.5, 8, containing final concentrations of 75 µM monensin, 10 µM nigericin, 10 mM 2-deoxyglucose, and 10 mM $NaN_3$. Monensin and nigericin are ionophores that carry protons across the plasma membrane, while the 2-deoxyglucose and $NaN_3$ deprive the cell from ATP and thus block energy-dependent pH maintenance. Each point represents data from 20 cells. (C) Data collected from 80 yeast cells during the process of replicative aging. The young group consists of data points from the first recorded ratio at time 0 hr, and the old group consists of the last recorded measurement before cell death. Colored crosses indicate the average and bold lines show the median, p=9.1E-18. (D) Single-cell profiles of the pH at different replicative ages. Cells are grouped according to lifespan (RLS <15, 15–20, 20–25, 25–30 and >30). Grey bar indicates the range of pH values measured in the first time point (young age) to illustrate that 36% of the cells remain within this range throughout their lifespan. (E) Cytosolic acidity (normalized ratiometric pHluorin read-out:$F_{475}/F_{390}$, top) and replicative age (bottom) as a function of time, for a cell that enters senescence at 26 hr (dashed blue line) after which a sharp acidification of the cytosol (dashed red line) follows. (F) The pH of single cells during aging plotted versus their age in hours. Each grey line represents a single cell. The orange line shows the average pH. The dark grey circles show the total proteome pI in aging estimated based on the PI and changes in abundance of 1229 proteins. Error bars represent the standard error of the mean (SEM).

The online version of this article includes the following source data and figure supplement(s) for figure 1:

**Source data 1.** Singe-cell measurements of pH in replicative aging yeast cells.
**Figure supplement 1.** Age-related changes in cytosolic pH are not predictive for lifespan.
**Figure supplement 2.** Representative single-cell trajectories with high time course resolution to pinpoint timing of senescence entry and sharp cytosolic acidification.
**Figure supplement 3.** Single-cell trajectories with high time course resolution, categorized in qualitative groups based on their senescence status.

senescence, where the cell cycle is significantly extended, but sporadic divisions occur. It is likely that these cells later on become post-mitotic, but this phase is not captured within the duration of the experiment. Throughout the experiment, cells with gradual entry into senescence exhibit smaller (1.6-fold increase, SD = 0.22) and a more gradual increase in cytosolic acidity (*Figure 1—figure supplement 2B*; *Figure 1—figure supplement 3*). Lastly, cells that actively divide throughout the duration of the experiment (14% of the total population) exhibit little changes in their cytosolic pH (1.4-fold increase, SD = 0.14) (*Figure 1—figure supplement 2C*; *Figure 1—figure supplement 3*). In all cells, the fold increase in cytosolic acidity with aging correlates to the time cells spend in their last division ($R^2$ = 0.6; *Figure 1—figure supplement 2D*). Overall, these data show that the drop in pH is posterior to the entry into a post-mitotic state and the degree of cytosolic acidification is dependent on the time spent in post-mitotic state.

In all cells, more than one measurement could be performed during the same cell cycle, if its duration lasted longer than 2 hr, allowing us to observe cell cycle-induced pH fluctuations. We find that there are fluctuations in the pHluorin read-out with the different cell cycle stages, but also find these to be minor and generally overpowered by the larger changes occurring after entry into senescence (*Figure 1—figure supplement 2A–C*). Considering that the majority of cells collected in the aged population are senescent cells, we conclude that the increased heterogeneity in the old group (*Figure 1C*) is dependent on the senescent status of the cells and not on the cell cycle stage at which the last measurement was performed.

We next asked how changes in cytosolic pH compare to those measured in the vacuole and cell cortex. Using the pHluorin2 (*Mahon, 2011*) and the same microfluidic design *Crane et al., 2014*, Chen and colleagues (*Chen et al., 2020*) measured the changes in vacuolar pH allowing comparison with our datasets (*Figure 1—figure supplement 2E*). From both studies, it is clear that the vacuole and cytosol change their pH in opposite directions during aging where the vacuole decreases in acidity and the cytosol becomes more acidic. An early life increase in cortical pH was reported previously (*Henderson et al., 2014*), which is the opposite of a decrease in the cytosolic pH observed in our long-term experiments. This apparent dichotomy prompted us to ask whether changes in the aging proteome can account for the distinct behavior of the cytosolic pH in relation to the vacuole and cell cortex. Indeed, in addition to the activity of proton pumps, the strong buffering capacity of metabolites and amino acid side chains contributes to pH homeostasis (*Moriyama et al., 1992*). Because the concentration of amino acids with a physiologically relevant pKa at protein surfaces is orders of magnitude larger than the concentration of free protons at pH 7 (protein concentration is in the mM-range while pH 7 corresponds to 60 nM $H^+$), the proteome

represents a buffer for changes in pH. Thus, we assessed the proteome isoelectric point (pI) during yeast replicative aging.

We utilized available datasets for protein abundance during aging (*Janssens et al., 2015*), predicted isoelectric point (Saccharomyces Genome Database, SGD), and protein copy number (*Ghaemmaghami et al., 2003*). We used data from 1229 proteins and corrected our calculations for relative protein abundance, thus weighing the proteome pI for the copy number of each protein. We found that in young cells, the proteome pI is 7.1, which accounts mostly for the cytosolic part of the cell, according to the Panther database for gene ontology. In cells aged for 60 hr (average replicative age ~22 divisions), the proteome pI reaches as low as 6.7 (*Figure 1F*). While this analysis carries uncertainties, for example related to protein pI predictions and age-related aggregate formation, it is striking that the proteome pI roughly follows the pH of the cytoplasm during aging. This suggests that additional to changes at the level of proton pumps, changes at the level of the buffering capacity of the proteome may underlie the decrease in cytosolic pH.

## Validation of FRET-based crowding probes during yeast aging

Because pH has previously been related to macromolecular organization (*Joyner et al., 2016*; *Munder et al., 2016*), we asked whether macromolecular crowding also changes in aging yeast cells. It is not trivial to quantitatively measure crowding in living organisms (*Rivas and Minton, 2016*), and most estimates of crowding were obtained from measuring dry cell mass (*Cayley et al., 1991*; *Zimmerman and Trach, 1991*), which does not necessarily reflect the actual in vivo levels of crowding. Additionally, crowding has not yet been addressed in aging. We previously developed a genetically encoded FRET-based probe that enables quantification of macromolecular crowding in vivo (*Boersma et al., 2015*). The sensor is genetically encoded, and it harbors a FRET pair, connected with a flexible linker. When placed in a crowded environment, the probe will obtain more compressed conformations, thus increasing the FRET efficiency. We utilized this FRET-based probe, named crGE, and harboring mCerulean3 as donor and mCitrine as acceptor (*Liu et al., 2018*). We also developed a new probe, named CrGE2.3 by exchanging the donor and acceptor for mEGFP and mScarlet-I, respectively (*Figure 2A*, *left*).

Through fluorescence microscopy, we determined the crowding read-out from the ratio of the intensity in the FRET channel and the donor channel (an example is shown in *Figure 2A*, *right*). When the crowding levels are higher, the fluorescence signal in the FRET channel also increases, leading to a higher FRET ratio and vice versa. To validate the read-out of the FRET probes, we induced a hyperosmotic shock by resuspending cells from an exponentially growing culture in 1M NaCl. Upon osmotic upshift, the water content and the cell volume should decrease, resulting in increased macromolecular crowding. Indeed, we observed a significant increase (p<0.001) in the crowding ratio of 18% from 0.54 to 0.64 from three independent experiments for crGE (*Figure 2B*, *left*). To determine whether in the new sensor, crGE2.3, functionality is retained after exchanging the fluorophores, we again induced hyperosmotic shock with 1M NaCl or 1.5M Sorbitol and observed significant increases in crowding ratio (p>0.001) of 23% from 1.19 to 1.54 for NaCl and to 1.46 for Sorbitol (*Figure 2B*, *right* and *Figure 2—figure supplement 1I*). These results show that both sensors are responsive to crowding changes in yeast in a reproducible manner.

In order to perform crowding measurements in aging cells, the sensor should be insensitive to age-related changes, such as proton concentration or fluorescent protein maturation. Because the cytosolic pH decreases with age, it could compromise read-outs from the crowding sensor. To estimate the pH sensitivity of the crGE and the crGE2.3 probes, we resuspended cells permeabilized with ionophores in buffers with pH ranging from 5 to 8. Our results show that read-outs from the sensor harboring the original FRET pair, mCerulean3-mCitrine, has a strong linear dependence on the intracellular pH (*Figure 2C*, *left*). When assessing the pH sensitivity of the new sensor, CrGE2.3, we found that this probe gives more stable read-outs within the pH range observed in aging (*Figure 2C*, *right*).

Fluorescent protein maturation also plays a role: Aging cells do not maintain the same division frequency, but transition to a slow and irregular division mode, called the senescence entry point (SEP) (*Figure 2—figure supplement 1*, *top row*) (*Fehrmann et al., 2013*). When the SEP occurs, the relative rates of sensor synthesis, degradation, and dilution through division are all disrupted, leading to an increase in fluorescence of the slower maturing fluorescent protein (*Figure 2—figure supplement 1A*). For aging studies, the FRET pair in crGE2.3, which has similar pH sensitivity and

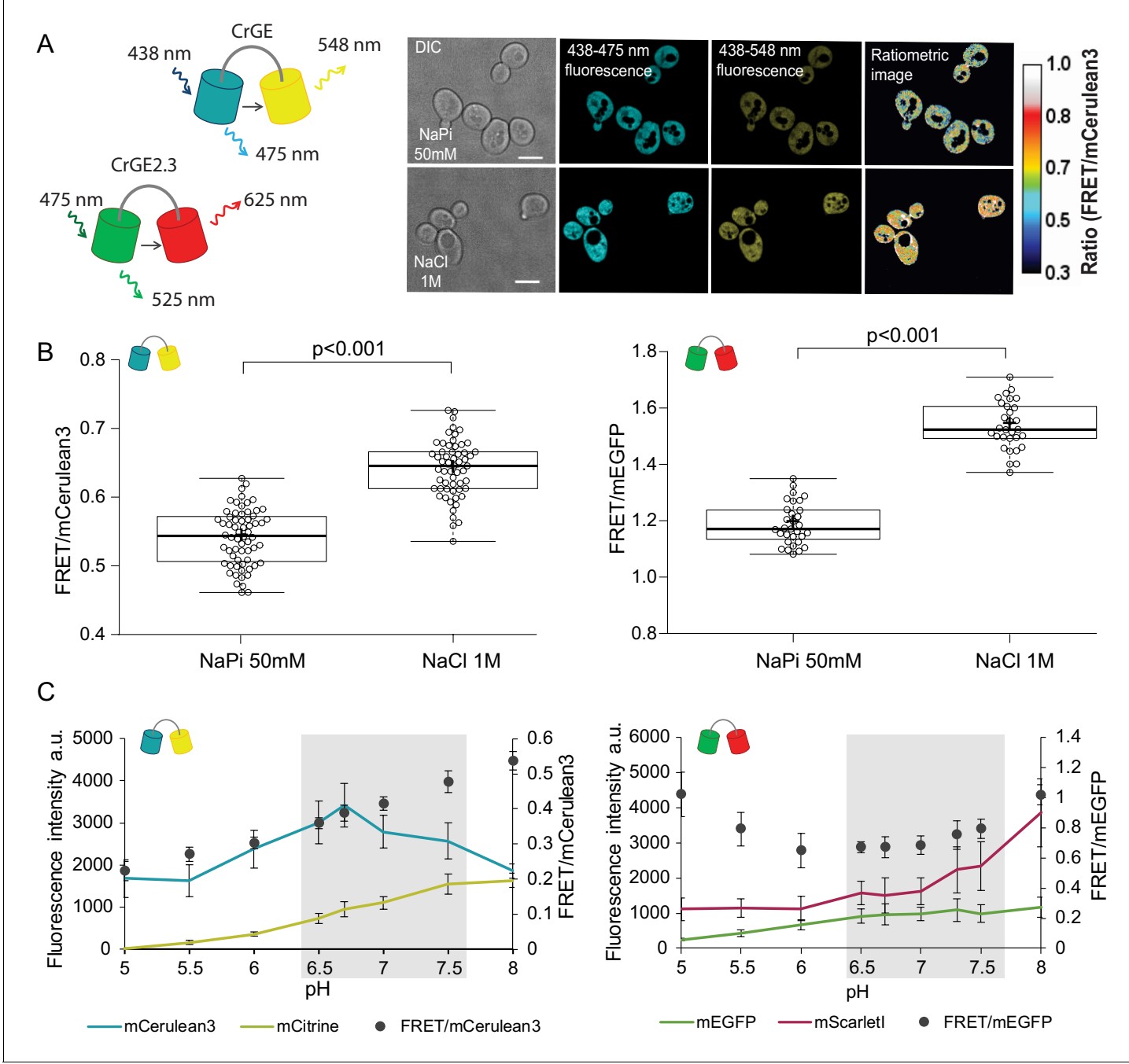

**Figure 2.** Crowding sensor with traditional CFP-YFP FRET pair is functional in yeast cells, but sensitive to pH variations; the sensor CrGE2.3 with mEGFP-mScarlet-I is aging-compatible. (**A**) Left: Schematic representation of the crGE and CrGE2.3 sensors. Right: Yeast cells expressing crGE sensor, the DIC, mCerulean3, and FRET channels are shown. Ratiometric image shows changes in the crowding ratio upon osmotic shock. The scale bar is 5 µm. (**B**) CrGE sensor (left), expressed under a strong constitutive TEF1 promoter in yeast, shows an increase in FRET/CFP ratio upon osmotic shock with 1M NaCl versus control in sodium phosphate buffer at pH 7. Crosses indicate the average and bold lines show the median. The data is from n > 58 cells per condition; p=2.0E-25. Right: CrGE2.3 sensor, expression and osmotic shock experiment is the same as for the original CrGE sensor, the data is from 30 cells per condition, p=1.6E-24 (**C**) Left: Fluorescence intensities of mCerulean3 (blue line), and mCitrine (yellow line; directly excited) and FRET/CFP ratios (black circles). Right: Fluorescent intensities from mEGFP (green line) and mScarlet-I (magenta; directly excited) and FRET/mEGFP ratio (black circles) from permeabilized cells, expressing the crowding sensor CrGE (left) and CrGE2.3 (right), in buffers with pH ranging from 5 to 8. Each data point represents the average of 20 cells; error bars show standard deviation.

The online version of this article includes the following source data and figure supplement(s) for figure 2:

**Source data 1.** Validation of FRET sensors GrGE and GrGE2.3.

*Figure 2 continued on next page*

*Figure 2 continued*

**Figure supplement 1.** Read-outs from fluorescent sensors are sensitive to conditions affecting the amount of actively fluorescent donor and acceptor of FRET, such as changes in division frequency and varying intracellular pH.

maturation kinetics between the two fluorescent proteins, is thus more suitable than the original sensor. To eliminate additional systematic errors, we corrected for an unequal number of donor and acceptor fluorophores by FRET normalization ($N_{FRET}$) (*Xia and Liu, 2001*) as pH-induced fluorescence quenching leads to a different number of fluorescent proteins and the same accounts for the proportion of fully matured sensors. Indeed, determining the $N_{FRET}$ eliminates differences between cells treated with 1 µM cycloheximide (*Figure 2—figure supplement 1A–D*) and in permeabilized cells at pH levels of 7 and 7.5, which are physiologically relevant pH levels where mScarlet-I signal shows pH dependence (*Figure 2—figure supplement 1E and F*). Additionally, the normalization retains read-outs from crowding changes (*Figure 2—figure supplement 1G–J*). Thus, in crGE2.3 we have increased the robustness of the probe to make it suitable for challenging long-term aging experiments.

## Crowding homeostasis is maintained during yeast replicative aging

To determine age-associated changes in macromolecular crowding, we constitutively expressed the optimized crowding probe in yeast cells immobilized in the ALCATRAS microfluidic device (*Crane et al., 2014*). As with the pH-sensor , we observe that the crowding probe localizes in the cytosol and nucleus, but it is excluded from other membrane-enclosed organelles (*Figure 3A*). We performed three independent experiments (*Figure 3B*, *Figure 3—figure supplement 1B and C*). From the first experiment, we collected data from 80 cells in the time course of 70 hrs and determined $N_{FRET}$ and FRET/mEGFP ratios (*Figure 3—figure supplement 1A*). Our measurements show heterogeneity in crowding levels in young cells and more so in old cells (*Figure 3B*, *Figure 3—figure supplement 1B and C*). Overall, we find that macromolecular crowding is maintained in the course of aging, with average ratios of 0.51 for both young and old cells, where, as in *Figure 1*, the 'young' group reflects the first and the 'old' group the last crowding measurements (*Figure 3B*). Two additional independent experiments showed similar trends in aging, albeit with small variation in absolute crowding levels (*Figure 3—figure supplement 1B and C*).

Plotting single-cell trajectories for cells that reach a replicative lifespan of 10, 10–15, 15–20, 20–25, or larger than 25 shows that the shortest-lived cells tend to increase the crowding levels during their lifespan, while the longer lived cells tend to have more stable crowding levels (*Figure 3C*). Indeed, there is a weak correlation ($R^2 = 0.14$, $p<0.001$) between lifespan and old age crowding levels (*Figure 3—figure supplement 1E*), and the fold change in $N_{FRET}$ ratios in aging shows a weak correlation with lifespan ($R^2 = 0.22$, $p<0.001$) (*Figure 3D*). In support of the relationship between crowding and aging, we observe that cells that live shorter than the average lifespan of 18 divisions have significantly higher ratios in aging ($p<0.01$), compared to long-lived cells (*Figure 3E*). It seems that it is the maintenance of crowding homeostasis, rather than the absolute crowding levels, which has an association with lifespan, as lifespan does not correlate to the crowding ratios in young cells (*Figure 3—figure supplement 1F*).

## Volume distribution of cellular compartments changes disproportionally in yeast replicative aging

Cell volume increases in aging and the increase per division is predictive for the lifespan of cells (*Janssens and Veenhoff, 2016b*). Because macromolecular crowding is directly linked to cell volume, we aimed to determine how the volume of the cytosol changes in aging and assessed aged cells on the ultrastructural level. To explore the ultrastructure of aged cells exclusively, we labeled the cell wall of cells in log-phase with Alexa-488, which were then allowed to age over 20 hr. As the dye remains with the aging mother cells (*Smeal et al., 1996*), we can specifically identify aged cells within the population using correlative light (*Figure 4A*) and electron microscopy (CLEM) (*Figure 4B*). Aged cells make up only a minor fraction of the population; at the 20 hr time point used here, each aged (and labeled) cell is outnumbered by approximately 8000 daughter cells. Although the exact age of each cell was not determined, based on population doubling times we

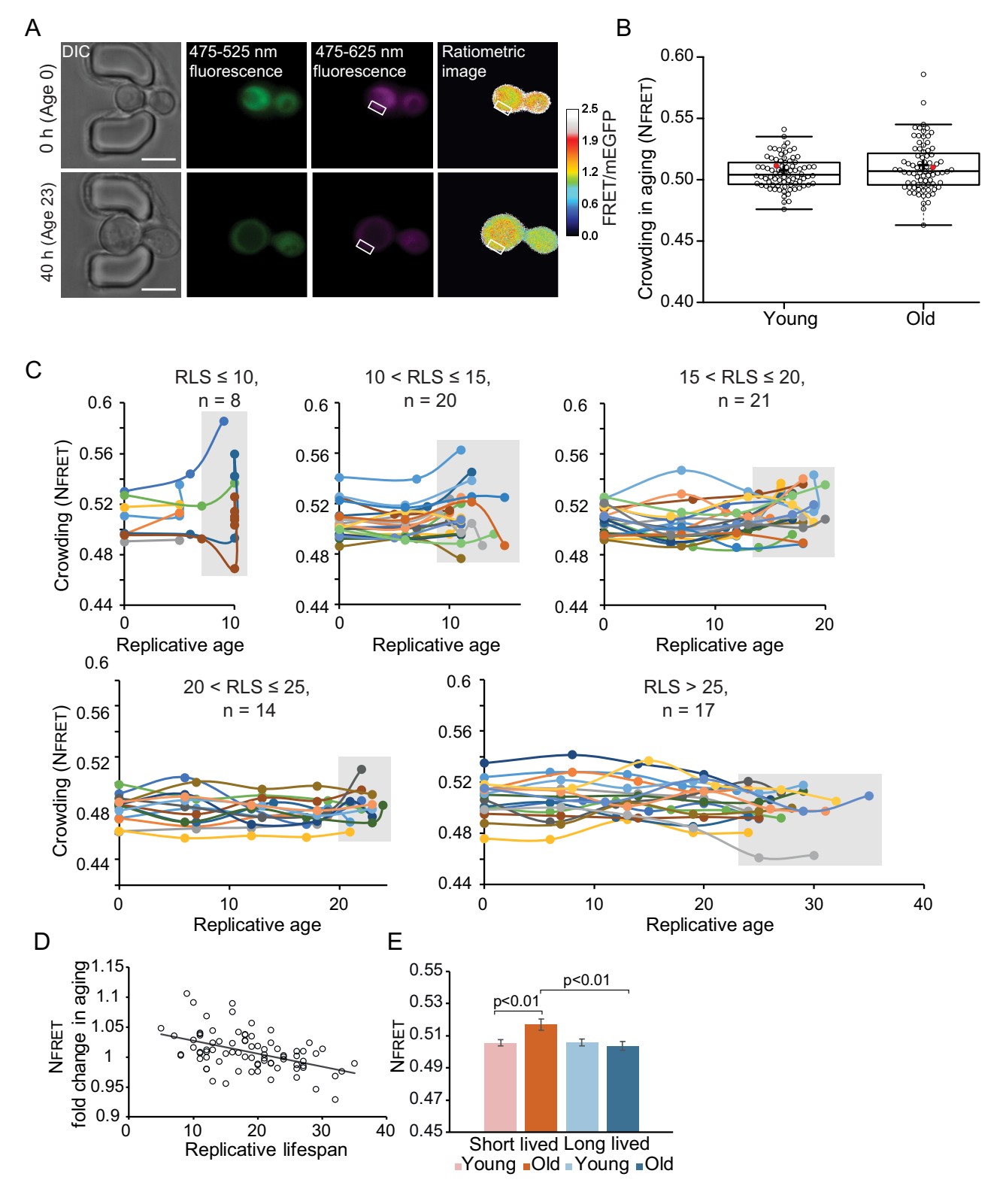

**Figure 3.** Crowding remains remarkably stable in aging. (**A**) Yeast cells expressing the crGE2.3 sensor, trapped in the aging chip. Images are from the same cell at the beginning of the experiment and the last measurement taken before cell death. Fluorescent images are taken once every 10 hr. Boxes indicate a cytoplasmic region. The scale bar is 5 μm. (**B**) Boxplot, showing normalized crowding ratios in young and old cells. For the young population, the first ratios recorded at the start of the experiment were taken. For the old population, the last ratio recorded before cell death was taken. The old

*Figure 3 continued*

population contains cells from different ages with a median lifespan of 18 divisions, n = 80. The ratio recorded for the cell displayed in *Figure 3—figure supplement 1A* is shown with a red dot. (C) Single-cell trajectories of cells with indicated replicative lifespan-ranges and an indicated number of cells in each category. Grey boxes indicate the range of crowding ratios at the end of the replicative lifespan (RLS) of each age group. (D) The fold change in crowding plotted against the replicative lifespan of cells, $R^2$ = 0.22, p=1.3E-05. (E) The average first and last recorded $N_{FRET}$ values of cells with an RLS shorter (red) or longer (blue) than the median RLS of 18 divisions. There is no difference in crowding between young and old cells from the long-lived population. However, in the short-lived population, the old cells have significantly higher crowding than the young cells (p=0.007), and the old cells from the long-lived population (p=0.003).

The online version of this article includes the following source data and figure supplement(s) for figure 3:

**Source data 1.** Singe-cell measurements of macromolecular crowding in replicative aging yeast cells.

**Figure supplement 1.** Crowding levels vary between individual young cells, but starting crowding levels do not associate with lifespan.

estimate that the Alexa-488 labeled cells will have performed on average 13 divisions, with a significant spread due to cell-to-cell differences in doubling time (*Janssens et al., 2015*). Fourteen tomograms of aged cells and 10 tomograms of young cells were acquired and segmented to define the plasma membrane, nucleus, vacuole, lipid droplets (LD), endoplasmic reticulum (ER), multivesicular bodies, and mitochondria (*Figure 4—figure supplements 1* and *2*). The aged cells show diverse phenotypes in terms of, *e.g.*, numbers and size of vacuoles, or numbers of lipid droplets. The lipid droplets become more prevalent in aged cells, confirming previous findings (*Beas et al., 2020*; *Figure 4—figure supplement 3*). In almost all cells, especially the vacuoles take up a larger fraction of the cell volume than they do in young cells.

To quantify organellar volume in the tomograms, we focused on the vacuole and nuclei, which are the largest compartments. We calculated their volume and membrane surface areas relative to the total cell volume and plasma membrane surface area in each tomogram (*Figure 4C*). We find that the vacuoles take up 3% to 24% v/v in young cells and a higher volume fraction, namely 16% to 66% v/v, in old cells. The nuclei take up between 8% and 37% v/v in young cells, which is similar to old cells (7–30% v/v). Interestingly, we find that in old cells, vacuoles can occupy up to 66 % v/v, this is comparable to the theoretical maximum volume of 64% v/v that randomly packed spheres can occupy in a container. Subtracting the nuclear and vacuolar volumes from the total volume provides an estimate of the cytosolic volume in the analyzed tomograms. The cytosol volume takes up 54–82% v/v in the young-cell sections compared with 22–70 % v/v in those from old cells. Consistent with the changes in organelle volumes in aging, we find that the surface area of the vacuole and nuclear membranes relates to the surface area of the plasma membrane in the young cell sections. In contrast, the relation between organellar and plasma membrane surface areas is lost upon aging (*Figure 4D*). The loss of this relationship provides the correspondingly wide distribution in occupied organellar volumes.

To assess the consequences of the changes in organellar crowding, we determined the average distance between organelles (*Figure 4E*), which would be a measure for cytoplasmic confinement of larger particles induced by organelles. We find a striking decrease in the most common distance from ~500 nm in young cells to ~100 nm in old cells, albeit with a wide distribution. Given a diffusion coefficient of 0.1 μm²/s previously determined in yeast cells for a particle with a diameter of 40 nm, it would take 0.005 s versus 0.3 s to reach a membrane with Brownian diffusion alone, assuming similar viscosity (Materials and methods; supplementary information, *Figure 4—figure supplement 4*).

Our analysis of the ultrastructure of aged cells further suggests that even though the total cell size increases in replicatively aging cells (*Crane et al., 2014*; *Fehrmann et al., 2013*; *Janssens and Veenhoff, 2016b*; *Lee et al., 2012*), the cytoplasmic volume fraction does not increase in aging, but rather remains stable or even decreases as the vacuolar volume fraction increases. Moreover, we conclude that crowding at the length scales of organelles, which we coin organellar crowding, strongly increases with aging.

## Discussion

Here, we provide an analysis of the progressive change during aging for several parameters that define a cell's intracellular environment; namely, cytosolic pH and crowding on the scale of macromolecules and organelles, all impinging on the hallmarks of aging. We find that the largest changes

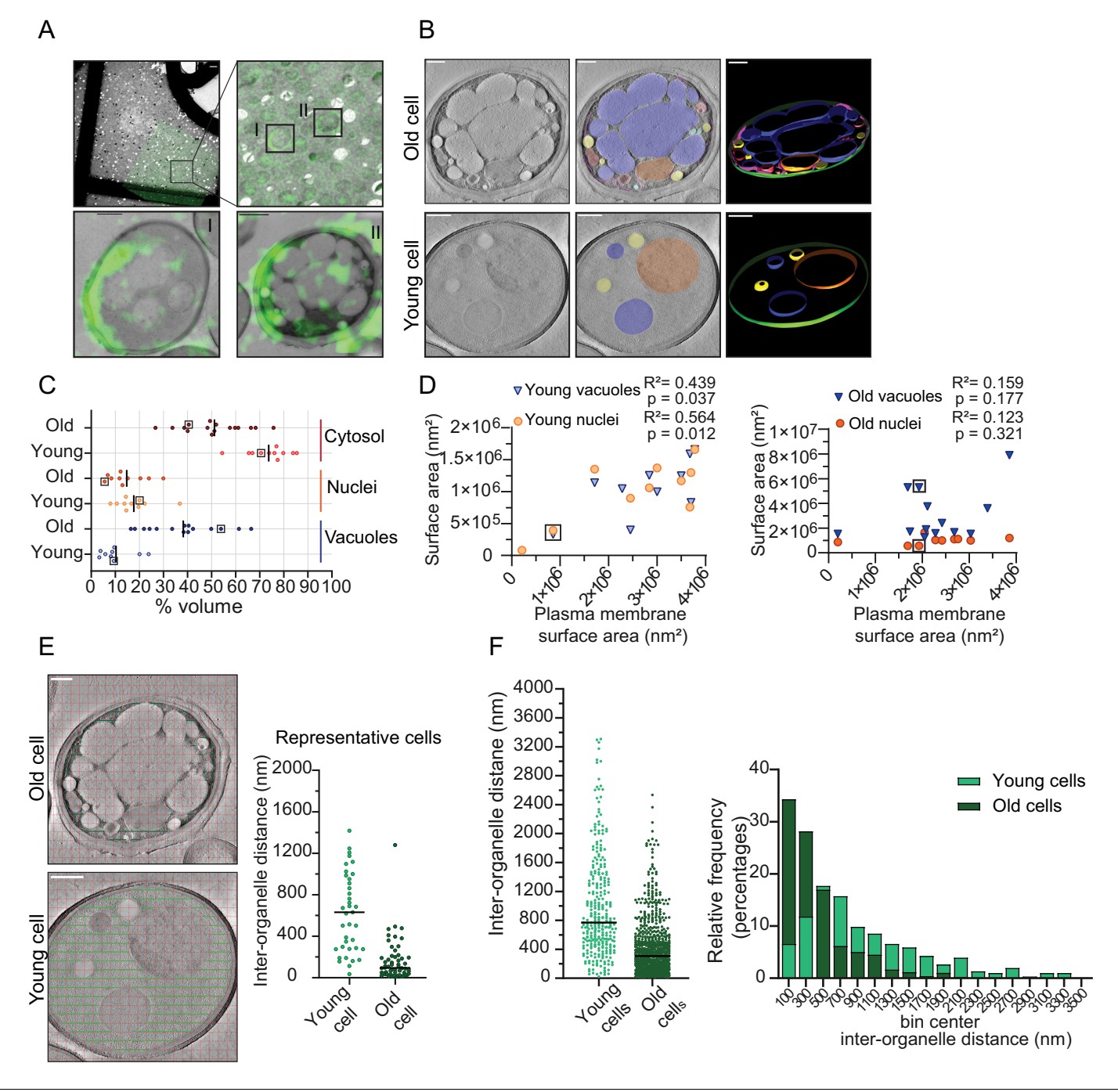

**Figure 4.** Inter-organellar crowding increases in aged cells. (**A**) Identification of aged cells (20 hr, replicative age ~13) by CLEM. Overlay of low magnification (225x) electron micrograph and fluorescence image and zoom-in of old cells with Alexa-488 labeling (boxed). Scale bar is 10 μm. Boxed cells I and II are shown at higher magnification (8900x) in lower panels with a scale bar set at 1 μm. (**B**) Single slices of tomograms without (left panel) and with an overlay to emphasize organelles (middle panel). 3D isosurface rendering (right panel) of tomograms of young or old cells. Nuclei (orange), vacuoles (blue), lipid droplets (yellow), mitochondria (red), ER (magenta), and plasma membrane (green). Scale bars are 500 nm. (**C**) The relative cell volume occupied by vacuoles increases in old cells. The scatterplot shows volume fractions of nuclei and vacuoles in 14 aged cells and 10 young cells. The volume fraction of the cell section minus the volume fractions of the nucleus and vacuole is used to estimate of the cytosol volume fraction. The values for the example shown in B are boxed. Black lines denote the median. (**D**) Membrane surface areas of nucleus and vacuole as a fraction of the plasma membrane surface area in young (left) and aged (right) cell sections. Each point represents a single nucleus or the sum of all vacuole surface areas within a single cell. The boxes indicate values from cells in B; linear correlations, $R^2$ values, are indicated. (**E**) Method to measure the distance between membranes in young and aged cells. Right is a scatter plot of measured distances from represented cells (**F**) Left is a scatter plot of inter-

*Figure 4 continued on next page*

*Figure 4 continued*

organelle distances between membrane-bound organelles in young and old cells. The black line demarks the median. Right is a histogram of inter-organelle distance distribution. Values were grouped into 200 nm width bins. n = 305 from 10 young cells, n = 778 from 14 old cells.

The online version of this article includes the following source data and figure supplement(s) for figure 4:

**Source data 1.** Volume, surface area, and interorganelle distance measurements in young and replicative aged yeast cells.
**Figure supplement 1.** 3D isosurface rendering of tomograms of 10 young cells.
**Figure supplement 2.** 3D isosurface rendering of tomograms of 14 aged cells (~age 13).
**Figure supplement 3.** Number and surface area of lipid droplets increase in yeast replicative aging.
**Figure supplement 4.** The time to diffuse between two membranes depends strongly on particle-size.

arise from organellar crowding, exemplified by the average membrane-to-membrane distance in aged cells being >2 times smaller in aged cells than in a young cell, while macromolecular crowding homeostasis is mostly maintained. The cytosolic pH shows a progressive decline that follows the pI of the proteome throughout much of the lifespan, while a steeper decrease in pH is observed at senescence. Below, we discuss possible causes of this changed intracellular environment and implications on molecular processes in an aging cell.

## pH homeostasis in aging

In general, pH has far-reaching consequences on cell physiology. Among others, the pH influences protein folding, enzyme activity, phosphorylation of metabolites and proteins, protein solubility/phase separation, interactions between the molecules, redox potential, proton gradients, and proton-dependent transport of nutrients (*Orij et al., 2011*). Even pH variations of 0.5 pH as we measure here, cause various enzymes to lose activity (*Ju et al., 2004*; *Talley and Alexov, 2010*) or induce liquid-liquid phase separations of proteins in cells (*Triandafillou et al., 2018*).

Because the drop in cytosolic pH has limited predictive value for lifespan (*Figure 1—figure supplement 1F*), we conclude that it is not a general early cause of aging. Significantly, we find that the sharp acidification of the cytosol only occurs after the cells become senescent. This suggests that senescence entry is not driven by cytosol acidification and there is possibly another underlying cause for both entry into senescence and cytosol acidification. Additionally, the degree of acidification of the cytosol, and therefore the increased heterogeneity in the old population, is related to the time spent in the senescent state. We speculate that a low availability of energy in the aging cell is a potential driver of senescence entry and cytosolic acidification.

Previously pH has been linked to aging in studies that showed both the vacuole and cell cortex become more basic with age (*Henderson et al., 2014*; *Hughes and Gottschling, 2012*) and the cytosol more acidic (*Knieß and Mayer, 2016*). Our findings add to this work, providing new insights only possible from the single-cell level. We show that cytosolic acidity strongly increases only after entry into senescence and we do not observe drastic changes in early lifespan. The changes in cytosolic pH are thus decoupled from those in the vacuole, which occur very early in the replicative lifespan (*Chen et al., 2020*; *Hughes and Gottschling, 2012*) and have no established connection to the timing of the senesce entry point. Comparing with previous measurements of the pH at the cell cortex with a plasma membrane-anchored pH sensor (*Henderson et al., 2014*), our measurements with a freely diffusing cytosolic pH sensor suggest that the plasma membrane sensor reports a local pH distinct from the cytosol. At the cortex, the pH was shown to alkalize already in age 3 cells, while the cytosolic pH acidifies mostly so after the cells enter senescence. Previously, an elegant model was proposed that explained why the pH of the vacuole and at the cell cortex alkalize in aging cells, which entailed competition for protons between the vacuolar and plasma membrane ATPases (*Henderson et al., 2014*). From the distinct timing of changes in vacuole and cytosol, plus the distinct directions of the changes in the cytosol and cell cortex, we conclude that this model alone cannot fully explain the acidification of the cytosol. We suggest that pH homeostasis in the cytosol relies on additional mechanisms, and present data to support that the proteome itself could provide buffering capacity (discussed below). To explain the alkalinization of the cell cortex in aging cells, we speculate that the high concentrations of proton pumps in the plasma membrane contribute to creating an alkaline microenvironment. On the other hand, the alkalization of the vacuoles could, besides the previously proposed competition for protons (*Henderson et al., 2014*), be driven by the

loss of functional vacuolar ATPases. The latter relation is supported by the observations of loss of the stoichiometry of the ATPase subunits in aging (*Janssens et al., 2015*) and the partial recovery of the stoichiometry after overexpression of the Vma1 component, which subsequently increases lifespan in yeast (*Hughes and Gottschling, 2012*). In future studies, the measurement of the pH at all three subcellular locations simultaneously in single cells and at high time resolution, combined with interventions at the proteome level plus the activities of the ATPase, would test this model.

Can the decreasing pI of the proteome, observed in aging be a source for the drop in cytosolic pH? Cellular proteins have weakly acidic or basic residues, which collectively act as a buffer of cellular pH. It is proposed that yeast proteins evolved to have a pI that is similar to the pH of the compartment where they usually reside in *Brett et al., 2006*, although this is not described in human cells (*Garcia-Moreno, 2009*). However, a cell must regulate its pH away from the pI to maintain protein solubility and prevent protein phase separation or aggregation. Thus, the yeast proteome may provide the basal pH of a cell compartment, while energy-dependent mechanisms regulate it away from the pI. This principle has been demonstrated in vitro, where isolated lysosomes maintain their acidity through a Donnan-type equilibrium (*Moriyama et al., 1992*). Hence, if a cell's energy state does not allow proper pH regulation, the pH would follow the pI. Therefore, the drop in pI of 7.1 to 6.7 could be responsible for the modest drop in pH during replicative aging, and when cellular energy is lost upon senescence, a more considerable drop ensues. In young cells, the vacuolar pH is around 5.5 (*Plant et al., 1999*), while its pI is 6.5 (*Brett et al., 2006*). Unfortunately, the vacuolar pH sensor read-outs were not calibrated to yield actual pH values (*Chen et al., 2020*; *Hughes and Gottschling, 2012*), but it would be interesting to know whether pH levels in the vacuoles also approach values equal to the pI, as in the cytosol. Such observations raise the question of whether in old cells, the proteome becomes not only the main buffering mechanism, but also a regulatory mechanism for cellular pH. The cell-to-cell variation in pH that we observe during aging could, in turn, reflect variation in proteome composition or energy state. Additional mechanisms could add to the changes that we observe, such as altered proton transport activity, accumulation of acidic metabolites, or polynucleotide content. Nevertheless, the qualitative similarities between the pI and pH are striking and suggest the proteome contributes to the cytosolic pH in aging.

## Crowding homeostasis in aging

Physicochemical parameters such as pH and macromolecular crowding can determine biochemical organization in yeast (*Joyner et al., 2016*; *Munder et al., 2016*) and potentially cause disease (*Patel et al., 2015*). Changes in physicochemical homeostasis could explain the multifactorial nature of the aging process since they will have ubiquitous and diverse effects on all cellular processes (*Alberti and Hyman, 2016*). We investigated macromolecular crowding as an intrinsic cell property, which can modulate, for example, phase separation and transition in vivo (*Delarue et al., 2018*; *Joyner et al., 2016*). Considering the significant increase in volume the cell undergoes with age, it is at first somewhat surprising that we find crowding levels to be very stable in old yeast cells. This stability may be a consequence of a combination of relatively small changes in cytoplasmic volume fraction (discussed below) and efficient mechanisms to maintain macromolecular crowding homeostasis.

To date, the only way to change crowding levels in the cells considerably and rapidly is to apply an osmotic upshift in the medium. The cell responds immediately by uptake of potassium ions to regain its volume and crowding (*Granados et al., 2018*). Even in a potassium-deprived medium, the crowding effects are observed for a short period due to other response mechanisms and/or efficient uptake of trace amounts of potassium. These short response times indicate that crowding levels are crucial. Crowding could be regulated by an array of mechanisms that prevent drift in crowding over more extended periods. These include, for example, (1) uptake of counter ions upon biopolymer synthesis, inducing an osmotic pressure over the membrane resulting in cell growth and reduction of the biopolymer concentration (*Basan et al., 2015*), (2) carbon catabolite repression to reduce the space taken up by metabolic enzymes (*Zhou et al., 2013*), or (3) altering the ribosome/tRNA concentration (*Delarue et al., 2018*; *Klumpp et al., 2013*). These, and yet to be identified mechanisms could regulate crowding in aging. The robust sensor developed here will provide a valuable tool to identify the genes that maintain macromolecular crowding at the nanometer scale.

Are there consequences of crowding during aging? We find that cells with crowding levels that remain relatively unchanged during aging have a longer lifespan, while cells increasing in crowding tend to be average or shorter-lived. Young cells display a natural variability of crowding between

cells that is independent of lifespan. Therefore, retaining initial crowding levels could be beneficial for old cells. A too-large drop in crowding would reduce cell viability: It was recently suggested that dilution of the cytoplasm could evoke cell cycle arrest and lead to senescence, through a variety of mechanisms (*Neurohr et al., 2019*). Possibly, an optimum macromolecular crowding exists, from both a physicochemical and biochemical viewpoint, within a window of less optimal but viable crowding levels.

## Organellar crowding in aging

Live-cell imaging studies of yeast expressing GFP-tagged organelle markers have highlighted how several organelles change in abundance or shape in aging yeast cells. Amongst them are the increase in vacuolar size (*Lee et al., 2012*) and fragmentation of nucleoli (*Crane et al., 2019*; *Kennedy et al., 1997*) and mitochondria (*Scheckhuber et al., 2007*). However, because aged cells are scarce in exponentially growing cultures, the detailed ultrastructural properties of aged cells had not yet been researched using EM analysis. We use CLEM to reveal that aged cells have an altered ultrastructure. Particularly striking is that the available space for the cytoplasm can become minimal in aged cells and is enclosed by a large surface area of organellar membranes. The average membrane-to-membrane distance in aged cells is >2 times smaller in aged cells than in a young cell: The average distance between organelles decreases from ~1000 to <500 nm. The distribution is, however, strongly tailing and these averages correspond to a change in the most common distance from 400–600 to 0–200 nm. Already from the frequency distribution of the measured inter-organelle distances that are smaller than 80 nm, which is ~12% in aged cells and ~2% in young cells, one could deduce that contact sites are possibly expanded in aged cells. The implications of an increase in membrane contact sites can be widespread. Moreover, the enlarged compartmentalization must affect the movement of larger structures such as ribosomes and induce confinement on similar-sized particles in the 40 nm range.

The organellar crowding should have several effects that are highly dependent on the local distance between the membranes and the size of the particle, as demonstrated by the time required to diffuse to the membrane (supporting information, *Figure 4—figure supplement 3*). The proximity to the membrane also increases the likelihood of interactions with membranes. The particles will also suffer an entropic cost by sacrificing translational degrees of freedom in the inter-organellar spaces. Hence, larger particles may be crowded out of regions with high organellar crowding leading to a size-dependent spatial sorting. On a technical note, given the extreme dependence on particle size, particles that are much smaller than the distance between the organelles would notice less of this confinement, which would include, for example, the macromolecular crowding sensor, which is polymer-like with a radius of ~5 nm. In contrast, fluorescent particles with radii of ~20 nm should experience confinement when present in the typical 100 nm confinements. Therefore, the behavior of such a particle will be dependent on where it is inside the cytoplasm. From a biological perspective, future studies should address how altered spatial sorting of large biological relevant particles, such as ribosomes, RNPs, or protein condensates or aggregates, would affect the physiology of aged cells.

## An integrated view of physicochemical homeostasis in aging

It is well established that aged cells are derailed in molecular aspects, such as the loss of protein homeostasis, genome stability or metabolic state, and that these molecular changes affect the cell's physiology. However, the identity of cells and organelles is not only defined on a molecular composition level but is also defined by physical and physicochemical properties. The data presented here highlight that these are also drifting in aging. First, the physical property of organelle size drifts in aging: where the cytosol in young cells occupies most volume, followed by the nucleus and vacuole (*Figure 4D left*), in old cells the order is opposite: here the vacuoles are largest, and the cytosol represents the smallest volume fraction. In addition, the physicochemical property of pH changes in aging: where the pH of the vacuoles is kept much lower than the cytosol in young cells, their values come closer together as the cytosol acidifies (this study and *Knieß and Mayer, 2016*) and the vacuole loses acidity in aged cells (*Chen et al., 2020*; *Hughes and Gottschling, 2012*). Therefore, both in terms of size and pH, the vacuole and cytosol lose aspects of their compartmental identity. Lastly, crowding on the scale of organelles, which we term organellar crowding, sharply increases with

aging. Organellar crowding likely influences phenomena that are on the 100 nm to μm length scale, such as long-range diffusion of larger particles and organelles, condensate formation, organellar shape, RNA translation, and cytoskeletal dynamics. Besides, the increased surface area presented by the organelles may give additional opportunity for adsorption or increased membrane contact sites. However, crowding at the length scale of a single protein, that is, ~10 nm, changes little, and these proteins do not experience a direct effect of organellar crowding. By analogy, an ant would not notice if there were a fence around a field, but an elephant does. We speculate that while in young cells, micrometer length structures are mostly hindered in their diffusion throughout the cytosol by cytoskeletal structures, in aged cells, the high occupancy of intracellular organellar membranes provides the major obstacle. In the context of these significant changes in the physical and physico-chemical properties of aged cells, it is remarkable that macromolecular crowding on the scale of single proteins is rather stable in aging, and instigates future studies to identify what regulates these crowding levels.

## Materials and methods

### Plasmid construction and yeast strains

All yeast strains (*Supplementary file 1*) were constructed in the BY4741 genetic background (*his3Δ1, leu2Δ0, met15Δ0, ura3Δ0*) (*Brachmann et al., 1998*) and were transformed with pRS303 yeast integrative plasmid, harboring the respective sensor gene with a *TEF1* promoter and *CYC1* terminator. A complete list of primers used to construct the strains can be found in *Supplementary file 2*.

To construct SMY008, the yeast codon-optimized gene of the crGE-NLS sequence was amplified from pYES2 vector (GeneArt, Invitrogen) together with pTEF1 and CYC1T by PCR with the forward primer F1_SM and reverse primer R1_SM, introducing SalI restriction site downstream of the terminator and removing the NLS localization signal. The sequence was subcloned into a pRS303 yeast integrative vector in between SpeI and SalI sites. The construct was then used to obtain chromosomal integration of the sensor sequence in the *HIS* locus.

For the generation of SMY015, the yeast codon-optimized mEGFP-crGE-mCherry (GeneArt, Invitrogen) was amplified using primer F3_SM to introduce a HindIII site and the primer R3_SM to introduce a stop codon and a downstream XbaI site. The PCR product was subcloned in pYES2-TEF1 between HindIII and XbaI. The resulting TEF1-mEGFP-crGE-mCherry construct was amplified with F1_SM an R1_SM and subcloned in pRS303, as described above. The gene encoding for Gamillus-crGE-mScarlet-I (GeneArt, Invitrogen) was amplified with primers F4_SM and R4_SM to introduce a stop codon, as well as XmaI and XbaI restriction sites after the mScarlet-I. The resulting PCR product was digested with NcoI and XmaI to isolate the mScarlet-I gene. The mCherry sequence in the pRS303-mEGFP-crGE-mCherry was then substituted with mScarlet-I, by subcloning between the NcoI and XmaI restriction sites. The resulting construct of mEGFP-crGE-mScarlet-I in pRS303 was used for chromosomal integration into the *HIS* locus of BY4741.

To construct SMY012, the pHluorin gene was amplified from pYES2-ACT1-pHluorin (*Diakov et al., 2013*) by PCR, using primers F2_SM, introducing HindIII restriction site and AAAAAA for enhanced expression in front of the start codon, and R2_SM, introducing a stop codon and XmaI and XbaI restriction sites. The PCR product was subcloned in the pYES2-TEF1 vector between HindIII and XbaI restriction sites. The TEF1-pHluorin-CYC1T construct was then amplified by PCR with primers F1_SM and R1_SM and integrated into the pRS303 with SpeI and SalI restriction sites. All pRS303 constructs were sequenced.

### Media and growth conditions

Yeast cells were grown at 30℃, 200 rpm in Synthetic Dropout medium without histidine (SD-his), supplemented with 2% (w/v) glucose. Cells from an overnight culture are diluted 100 × in 10 mL of SD-his, 2% glucose. After 7 hr of incubation, appropriate dilutions were made to obtain cultures in the exponential growth phase on the following day ($OD_{600}$ = 0.4–0.7).

## Microscopy

All in vivo experiments were performed at 30˚C. Images were acquired using a DeltaVision Elite imaging system (Applied Precision (GE), Issaquah, WA, USA) composed of an inverted microscope (IX-71; Olympus) equipped with a UPlanSApo 100× (1.4 NA) oil immersion objective, InsightSSI solid-state illumination, ultimate focus, and a PCO sCMOS camera.

Excitation and emission were measured with the following filter sets, respectively, and in the indicated order: crGE: CFP 438/24 and 475/24 nm, YFP 513/17 and 548/22 nm, FRET 438/24 nm and 548/22 nm. crGE2.3: FITC: 475/28 and 525/48 nm, A594: 575/25 and 625/45 nm, FRET: 475/28 and 625/45. pHluorin: DAPI: 390/18 and 435/48 nm, FITC: 475/28 and 525/48 nm. For crGE and pHluorin, 32% transmission power and for crGE2.3 2% for the FITC channel and 32% for the A595 and the FRET channel. For the aging experiments, stacks of 3 or 4 images with 0.7 µm spacing were taken, and for other experiments, stacks of 30 images with 0.2 µm spacing were taken at an exposure time of 25 ms for all experiments.

## Aging experiments

The microfluidic chips were used as described previously (*Crane et al., 2014*). DIC images of the cells were taken every 20 min to follow the number of divisions for each cell and determine the replicative lifespan. Fluorescent images were collected every 10 hr. Z-sections were taken in both DIC and fluorescence imaging with three or four slices of 0.7 µm thickness. The experiment was left to continue up to 80 hr and only cells that were trapped from the beginning of the experiment and died in the device within the time course of the experiment were included in the analysis.

## Tracing of cytosolic acidity at high time frequency

Images were acquired using a DeltaVision Elite imaging system (Applied Precision (GE), Issaquah, WA, USA) composed of an inverted microscope (IX-71; Olympus) equipped with a UPlanSApo 100× (1.4 NA) oil immersion objective, InsightSSITM Solid State Illumination, excitation and emission filters DAPI: 390/18 and 435/48 nm, FITC: 475/28 and 525/48 nm, ultimate focus and a CoolSNAP HQ2 camera (Photometrics, Tucson, AZ, USA). Exposure time was 0.1 s at 32% transmission. Fluorescent images were collected every 1 hr and DIC images were collected every 20 min for the whole duration of the experiment of 50 hr. Only cells that were present in the microfluidic device from the beginning of the experiment were analyzed. All cells were analyzed regardless of whether they died, became senescent, or were mitotically active until the end of the experiment. After entering senescence, some cells begin to lose their fluorescence signal reaching levels close to those of the background. In these cases, the measurements were further processed only if the fluorescent signals from both channels were at least twice the background to ensure reliable read-outs from the pHluorin.

## Image analysis

Processing of all images was performed using Fiji (ImageJ, National Institutes of Health). For each image, the z-stack with the best focus was selected. pHluorin and the crowding sensors localize in the cytosol and nucleus, and appear to be excluded from the vacuole and probably also from other membrane-enclosed cytoplasmic organelles. We determined the fluorescence in each channel for the entire cell and subtracted the background from a region outside the cell. The respective ratios were subsequently calculated.

## Determining $N_{FRET}$

Fluorescence signals from the donor ($I_{mEGFP}$), acceptor ($I_{mScarletI}$), and FRET ($I_{FRET}$) channels after background subtraction were used to calculate the normalized FRET ($N_{FRET}$) (*Xia and Liu, 2001*). We did not correct for the donor bleedthrough in the FRET channel and the acceptor cross-excitation because these contributions were minimal.

$$N_{FRET} = \frac{I_{FRET}}{\sqrt{I_{donor} \times I_{acceptor}}}$$

## Statistical analysis

Statistical parameters, including the number of cells analyzed, are reported in the figures and corresponding figure legends. The significance of changes was determined with a two-tailed Student's t-test; linear regression analysis ($R^2$) was done in Excel, and Spearman's rank correlation coefficient ($r_s$) were calculated in Matlab.

## Relation between pHluorin ratios and pH values

As described in *Munder et al., 2016*, 2 mL of exponentially growing culture with $OD_{600}$ of 0.5 were centrifuged at room temperature for 3 min at 3,000 g in a tabletop centrifuge. The cells were then resuspended in 200 µL calibration buffer (50 mM MES, 50 mM HEPES, 50 mM KCl, 50 mM NaCl, 200 mM $NH_4CH_3CO_2$) at pH 5, 5.5, 6, 6.5, 7, 7.5, and 8. 75 µM monensin, 10 µM nigericin, 10 mM 2-deoxyglucose, and 10 mM $NaN_3$ (final concentrations) to each buffer. The cells were then loaded in microfluidic chips (see below), and their fluorescence was determined (*Figure 1B*).

## pH sensitivity of crGE and crGE2.3 crowding sensors

Cells from exponentially growing cultures at $OD_{600}$ of 0.5 were harvested and resuspended in the same calibration buffers titrated to pH of 5, 5.5, 6, 6.5, 6.7, 7, 7.3, 7.5, 8 as the pHluorin calibration. The FRET/CFP and FRET/mEGFP ratios were determined from cells on a glass slide.

## Cycloheximide treatment

A 20 mL exponentially growing culture was split into two cultures of 10 mL. To one of the flasks, a final concentration 1 µM cycloheximide was added from a 1000 × stock solution in DMSO. As a control, 10 µL DMSO was added to the other culture. Samples were collected immediately after the addition of cycloheximide or DMSO and imaged. Both cultures were then incubated at 30°C, shaking at 200 rpm for 90 min. Samples were collected after 90 min from both treatment and control cultures and imaged as described before.

## Osmotic shock

2 mL of exponentially growing culture was collected by centrifugation at 3000 g for 3 min. The cells were resuspended in 200 µL low osmolality buffer (50 mM NaPi, pH 7 for isotonic conditions) or high osmolality buffer (50 mM NaPi, 1 M NaCl or 1.5 M Sorbitol) to induce the osmotic upshift. Cells were then placed immediately on the glass slide and imaged.

## Proteome isoelectric point

To calculate the overall isoelectric point of the aging proteome, we used published datasets for the age-related change in protein abundance (*Janssens et al., 2015*), the protein copy numbers in young cells (*Ghaemmaghami et al., 2003*) and the computed isoelectric points (pI) (Saccharomyces Genome Database, SGD, https://www.yeastgenome.org/). We excluded 264 proteins from the aging dataset for which a copy number was not available. Out of the remaining 1229 proteins, 1071 proteins belong to the GO term 'cytoplasmic component' (Panther Gene Ontology; www.pantherdb. org), indicating the aging proteome mostly reflects (highly abundant) cytosolic proteins. For each time point in aging, the contribution of and individual protein to the overall pI was calculated by multiplying its pI with its relative abundance in the proteome. The total proteome pI was then derived as a sum of all weighed isoelectric points of the proteins in the dataset.

## Ultrastructural analysis of aged cells

WT cells were grown to mid log phase in SD medium supplemented with 1% glucose. $8 \times 10^7$ cells were collected by centrifugation, washed twice with PBS, and resuspended in 0.5 mL of PBS. 4 mg of EZ-link Sulfo-NHS-LC-Biotin (ThermoFisher Sci.) was dissolved in ice-cold $H_2O$ and immediately mixed with the cell slurry, which was incubated for 20 min at RT. Biotin-labeled cells were collected by centrifugation and washed twice with PBS. After resuspension in 100 µL of PBS, 5 µL of 5 mg/mL streptavidin-labeled with Alexa Fluor 488 (ThermoFisher Sci.) was added for 30 min. After washing in PBS, 40,000 cells were used to inoculate 30 mL of SD with 1% glucose, which was grown to an average age of 13 before processing for CLEM as follows. Cultures were concentrated into a thick slurry by gentle centrifugation and aspiration of excess media. These samples were high-pressure frozen in

a Leica HMP100, freeze-substituted in a Leica Freeze AFS with 0.1% uranyl acetate in dry acetone, and infiltrated with Lowicryl HM20 resin. The polymerized resin block was cut into ~200 nm thick sections onto 135 mesh H15 patterned copper/rhodium grids (Labtech). Fluorescence imaging was carried out as previously described in *Kukulski et al., 2011*. Fluorescent micrographs were acquired using a DeltaVision widefield microscope (Applied Precision/GE Healthcare) equipped with UPlanSApo 60x (1.64 NA) and 100x (1.4 NA) oil immersion objectives (Olympus), solid-state illumination and CoolSnapHQ$^2$ CCD camera (Photometrics). Bright-field and fluorescent images of grid squares with cells were acquired at both 60x and 100x magnification to facilitate alignment to electron micrographs in subsequent steps.

Grids were post-stained with lead citrate and labeled on both faces with 15 nm gold fiducials. Tilt series from $-60°$ to $60°$ of selected cells were acquired with a magnification of either 8900x or 13300x on an FEI F20 fitted with an FEI Eagle CCD camera (4k $\times$ 4 k) and using Serial EM (*Mastronarde, 2005*). Tomograms were reconstructed in IMOD (*Kremer et al., 1996*) using 15 nm gold fiducials for alignment. Low magnification (225x-440x) electron micrographs were also acquired to facilitate alignment with fluorescence micrographs and tomograms.

Correlation between light micrographs and electron micrographs was completed in Icy (*de Chaumont et al., 2012*) using the ec-CLEM plugin (*Paul-Gilloteaux et al., 2017*). Corresponding points in EM and light microscopy images were selected based on cellular features distinguishable in both the light microscopy and EM.

## Analysis of electron tomograms

Manual segmentation of tomograms was performed in IMOD/3DMOD (Version 4.9.8, *Kremer et al., 1996*) with contours drawn every 30 nm or less in z. The surface area and organelle volume were calculated from uncapped meshed models in IMOD. The cytosolic volume was estimated by subtracting the calculated volume of nuclei and vacuoles from the volume encapsulated by the plasma membrane.

Inter-organelle distances were calculated in an unbiased manner by overlaying horizontal lines spaced every 200 nm over the midplane of the reconstructed tomogram using the stereology tool in IMOD. Distances between organelles on overlaid lines were then drawn as contours and measured. As cortical ER was not faithfully visible in each cell, all ER membranes were excluded from the analysis. Graphs were compiled in Prism (GraphPad). The linear correlation coefficients $R^2$ were calculated in Prism (GraphPad).

## Dependence of particle size on diffusion time to a membrane

The weighted mean distance $d$ a particle travels by Brownian diffusion depends on the diffusion coefficient $D$ and the time $t$ following the equation (*Phillips et al., 2012*):

$$d = \sqrt{2nDt} \tag{1}$$

Here $n$ is the dimensionality, which we set to 1. In reality, the particle can move in 3 dimensions. We set $d$ as half the distance between the organelles, which is the distance the particle has to travel to the walls. The resulting $t$ provides the time a particle diffuses on average to hit the wall. We thus assume the particle starts in the middle, and therefore the time estimates are longer than when it would start at another point.

This formula is derived from the chance $c$ a particle has traveled to position $x$, which is at the organelle:

$$c(x,t) = \frac{N}{\sqrt{4\pi Dt}} e^{-\frac{x^2}{4Dt}} \tag{2}$$

The chance can be set arbitrarily to determine the time it requires for a particle to interact with the organelle. This formula will result in the same trends as *Equation 1* for observing the relative effect of the change in organelle distance upon aging. We, therefore, continue with *Equation 1*.

We take the distance the particle has to travel as $d = L/2$, with $L$ the distance between the organelles. To take into account the size of the diffusing particle, we set $d = L/2$ r, with $r$ = radius of the particle. Inserting into *Equation 1* gives for the average time for a particle to diffuse to the organelle:

$$t = \frac{\left(\frac{L}{2} - r\right)2}{2D} \qquad (3)$$

To compare old cells with young cells, we take the ratio $t$(young)/$t$(old) that provides the fold-decrease in time required to diffuse to the organelle membrane.

## Acknowledgements

We acknowledge The Netherlands Organization for Scientific Research: Vidi grant 723.015.002 to AJB and BBoL grant 737.016.016 to LMV, and the NIH RO1 GM105672 to DJT and CPL, and R01 AG056359, and P30 AG013280 to MK for financial support.

## Additional information

### Competing interests
Matt Kaeberlein: Reviewing editor, *eLife*. The other authors declare that no competing interests exist.

### Funding

| Funder | Grant reference number | Author |
|---|---|---|
| Nederlandse Organisatie voor Wetenschappelijk Onderzoek | 737.016.016 | Liesbeth M Veenhoff |
| Nederlandse Organisatie voor Wetenschappelijk Onderzoek | 723.015.002 | Arnold J Boersma |
| National Institutes of Health | RO1 GM105672 | C Patrick Lusk |
| National Institutes of Health | P30 AG013280 | Matt Kaeberlein |
| National Institutes of Health | R01 AG056359 | Matt Kaeberlein |

The funders had no role in study design, data collection and interpretation, or the decision to submit the work for publication.

### Author contributions

Sara N Mouton, Conceptualization, Formal analysis, Investigation, Writing - original draft, Writing - review and editing; David J Thaller, Formal analysis, Investigation; Matthew M Crane, Irina L Rempel, Owen T Terpstra, Anton Steen, Methodology; Matt Kaeberlein, C Patrick Lusk, Supervision; Arnold J Boersma, Liesbeth M Veenhoff, Conceptualization, Supervision, Writing - original draft, Writing - review and editing

### Author ORCIDs

Sara N Mouton https://orcid.org/0000-0001-9429-3788
David J Thaller http://orcid.org/0000-0003-3577-5562
Matthew M Crane http://orcid.org/0000-0002-6234-0954
Irina L Rempel https://orcid.org/0000-0003-4655-5311
Owen T Terpstra https://orcid.org/0000-0002-8767-4061
Anton Steen https://orcid.org/0000-0002-1064-6038
Matt Kaeberlein http://orcid.org/0000-0002-1311-3421
C Patrick Lusk http://orcid.org/0000-0003-4703-0533
Arnold J Boersma https://orcid.org/0000-0002-3714-5938
Liesbeth M Veenhoff https://orcid.org/0000-0002-0158-4728

### Decision letter and Author response
Decision letter https://doi.org/10.7554/eLife.54707.sa1
Author response https://doi.org/10.7554/eLife.54707.sa2

## Additional files

### Supplementary files
- Supplementary file 1. Strains used in this study.
- Supplementary file 2. Primer sequences used in this study.
- Supplementary file 3. Comparison between measuring cytoplasmic acidity in aging every 10 hr (*Figure 1C,D,F*, *Figure 1—figure supplement 1*) or every hour (*Figure 1E*, *Figure 1—figure supplement 2*).
- Transparent reporting form

### Data availability
All data generated or analysed during this study are included in the manuscript and supporting files. Source data files have been provided for Figures 1,2,3,4.

The following previously published dataset was used:

| Author(s) | Year | Dataset title | Dataset URL | Database and Identifier |
|---|---|---|---|---|
| Janssens | 2015 | Aging Yeast - Protein biogenesis machinery is a driver of replicative aging in yeast | https://www.ebi.ac.uk/pride/archive/projects/PXD001714 | PRIDE, PXD001714 |

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
