## [Decision Letter]

**Acceptance summary:**

As a Tools and Resources paper, this study provides a novel set of experimental data on the aspects of yeast aging that are not previously investigated. These datasets include cytosolic pH measurements, cellular crowding levels, and organellar volumes in replicatively aged yeast cells. Reviewers believe that these new datasets will provide an initial framework and important technical resources/methodology that could significantly accelerate studies addressing heterogeneity of the aging process.

**Decision letter after peer review:**

[Editors’ note: the authors submitted for reconsideration following the decision after peer review. What follows is the decision letter after the first round of review.]

Thank you for submitting your work entitled "A physicochemical roadmap of yeast replicative aging" for consideration by *eLife*. Your article has been reviewed by three peer reviewers, one of whom is a member of our Board of Reviewing Editors, and the evaluation has been overseen by a Senior Editor. The following individual involved in review of your submission has agreed to reveal their identity: Vyacheslav M Labunskyy (Reviewer #3).

Our decision has been reached after consultation between the reviewers. Based on these discussions and the individual reviews below, we regret to inform you that your work will not be considered further for publication in *eLife*.

Although reviewers have pointed out some novel aspects of this manuscript, especially reporting the pH and crowding measurements in the context of aging, during discussion the reviewers expressed a consensus view that (1) the manuscript in its current form lacks significant novelty for publication in *eLife*; (2) there are some common major concerns raised by reviewers, especially regarding the rigor and robustness of the measurements due to low cell numbers in many analyses; and (3) as a Tools and Resources paper, the manuscript should include a more comprehensive set of data measurements, especially for the CLEM data. The reviewers concluded that a significant amount of additional work will be needed in order to address all the major concerns raised by the reviewers, which lead to the decision to reject your submission. However, if you feel that these concerns can be effectively addressed in a major revision, we welcome an appeal detailing your plans.

Reviewer #1:

In the manuscript titled "A physicochemical roadmap of yeast replicative aging", authors Mouton et al. measured cytosolic pH, cellular crowding levels, and organellar volume in replicatively aged yeast cells, compared to young ones. Using a novel cellular pH sensor, the authors reported reduced pH in the cytosol in aged cells. Using a set of FRET-based crowding probes, the authors found no apparent change in cellular macromolecular crowding during aging. Using correlative light and electron microscopy, the authors measured the volume of vacuole and nuclei. This analysis showed both increased volume for vacuole and nuclei and increased inter-organellar crowding.

Overall, the authors used novel technologies and measured a number of cellular physiochemical changes during yeast replicative aging, providing a set of experimental data that are not previously available. However, there are a few key weaknesses even when considering this manuscript as a Tools and Resources paper without investigating in the mechanistic details.

1) Several assays only measured a very limited number of cells, especially for the cytosolic pH, where only 30 cells were analyzed. Importantly, as the authors noted, around 50% of these cells were able to maintain a pH in the normal range of young cells. This percentage appears to be even higher for cells that have relatively longer replicative lifespan. Given the broad variation in this phenotype, more cells should be included in this analysis.

2) Vacuole acidity is much reduced in aged cells due to imbalance in proton pumps between vacuolar membranes and plasma membranes (Hughes and Gottschling, 2012). Could this imbalance be the cause of cytosol acidification? Does cytosol acidification and vacuole deacidification concur at the single-cell level during aging? VMA1 overexpression has been shown to at least partially correct the proton pump imbalance resulting in extended lifespan. Does it also ameliorate the cytosol acidification during aging? Many of these questions can be experimentally tested.

3) CLEM analysis is another assay that was only performed for a very small number of cells (14 aged cells and 10 young cells). Although hundreds of section images were analyzed, they still only represent a very few selected cells. Given the general huge variation in cellular morphology, these small numbers are not sufficient to make a convincing case.

4) For the CLEM experiment, the authors mentioned that the cells were aged "over 20 hours to achieve an estimated replicative age of 13 divisions". Since these cells were imaged by fluorescent microscopy, would it be possible to precisely determine their replicative age by staining and counting bud scars? It seems to make sense to count bud scars for these 14 old cells and 10 young cells.

5) The authors stated that "14 tomograms were acquired and segmented to define the plasma membrane, nucleus, vacuole, lipid droplets, ER, multivesicular bodies, and mitochondria". Since these fine cellular structures, other than vacuole and nucleus, are clearly defined, why didn't the authors measure and compare them? Age-associated morphological changes to plasma membrane, lipid droplets, ER, vesicles, and mitochondria at the EM resolution are likely provide more insights to the physiology state of aging cells and support their conclusions based on the analysis of vacuole and nucleus.

Reviewer #2:

In this study, Mouton et al. use longitudinal tracking of single dividing yeast cell to follow the dynamics of several physiological markers using fluorescence microscopy in aging cells. Using the ratiometric probe phluorin, they report a gradual loss of pH in aging cells, that is not predictive of cell death. Then, they use a modified version of a molecular crowding sensor and report that crowding homeostasis is maintained during aging. Last, they perform cellular EM tomographic analyses to monitor the evolution of organelles size with age and they show that inter-organelles crowding increases with age.

Major concerns:

– The message conveyed by the paper is somewhat blurry, knowing that pH measurements do not appear to be conclusive: the authors confirm previous results showing a decrease in pH with age, but they show that it does not predict the evolution of cell fate. Also, they do not provide evidence that this observation is connected to any other well-described physiological marks of age (e.g. protein aggregates, ERCs, etc.). Similarly, they show that molecular crowding does not seem to be a marker of replicative aging. Therefore, even though some observations are novel in the context of aging (measurements of molecular and organelles crowding), the paper is lacking insights of high biological significance regarding the mechanism that controls the entry into replicative senescence.

– One weakness of the paper is that it appears as an assembly of disparate observations that are wrapped as a “physicochemical roadmap”. This term sounds quite obscure, and a bit oversold, or sophistic, knowing that it is only supported by a limited set of readouts that provide mostly inconclusive results.

– pH measurements in aging cells have already performed, leading to the same conclusion that pH decreases with age (using population-based measurements, Kniess et al., 2016, as acknowledged by the authors). However, another study (Henderson et al., 2014, also cited in the present manuscript) also reported measurements of “cortex-proximal cytosolic pH” in aging cells. In that paper, an increase in pH was observed. Whether the cortex-proximal cytosol has an identical pH as the “overall” cytosol is unclear at this point.

Therefore, knowing that the main merit of the paper is to attempt to confirm previous measurements of cytosolic pH, one would expect the authors to clearly address this controversy using in-depth complementary experiments. Instead, interpretations of results are made in such a way to give the impression that the results obtained by the authors is in line with all the literature, which is misleading.

In this context, this statement in the Discussion: “Here, we provide the first (to our knowledge) roadmap for the progressive decline in aging in several parameters defining a cell's intracellular environment, namely, pH, crowding, and volume, all of which impinge on the hallmarks of aging” is inappropriate, knowing that both pH and volume have already been documented in aging cells.

Reviewer #3:

In this manuscript, Mouton et al. have used a combination of single cell imaging and electron microscopy to survey heterogeneity in age-dependent physiological changes, including cytosolic pH, macromolecular crowding, and organellar size, during replicative aging in yeast. This manuscript addresses an important and interesting topic that has not yet been thoroughly investigated by previous studies. The authors made an interesting observation that in replicatively-aged yeast cytosolic pH gradually decreases early in life, but undergoes a significant drop when the cells stop dividing. However, at the single cell level old cells demonstrate incomplete penetrance phenotype, with only ~ 50% of the old cells showing this significant pH drop. Strikingly, despite significant increase in cell volume with age and increased organellar density, which authors refer to as "organellar crowding", macromolecular crowding remained very stable through the cell lifespan. This study is rigorously designed and of good technical quality. Although the exact mechanism for the incomplete penetrance remains unclear, this paper reports an initial framework and important technical resources/methodology that could significantly accelerate studies addressing heterogeneity of the aging process. I would support publication of the manuscript in *eLife* as Tools and Resources article. However, several points need to be further clarified by the authors.

1) One of the concerns is the low number of cells that were used for analysis of the pH and FRET reporters using microfluidics. Several hundreds of cells can be typically trapped and monitored though their replicative lifespan in the microfluidic device. Were any of the cells excluded from the analysis? If yes, what were the criteria for exclusion? An increase in the number of cells analyzed would help to better assess the proportion of cells that experience a drop in pH and strengthen their conclusions.

2) Another limitation of this study is that time intervals selected to collect fluorescent images for aging experiments (every 10 hours) do not allow assessing short-term physiological fluctuations. For example, does pH change in individual cells between different phases of the cell cycle? This issue needs at least to be discussed.

3) The Introduction section could be improved by logically putting the findings of this paper in the context of the published studies. Does the cell size (organellar volume) affect intracellular (organellar pH)? In several places, authors' statements are not supported by proper citations. E.g.: "In healthy young yeast cells, a "volume hierarchy" is observed where the cytosol represents the largest compartment, followed by the nucleus and vacuole." And "V-ATPase pumps protons from the cytosol into the lumen of various organelles and regulates their pH".

4) Another issue is related to the interpretation of the results. Although the data presented in this manuscript is not sufficient to assess pH changes in different cellular compartments (only in cytosol), authors should discuss whether the loss of vacuolar acidity with aging (Hughes and Gottschling, 2012) can potentially precede and/or be causative for the observed drop in cytosolic pH in old cells. Moreover, the decrease of cytosolic pH with aging (Knieß and Mayer, 2016) seems to contradict with previously reported data showing increased pH in the cell cortex of old mother cells (Henderson et al., 2014). What is the author interpretation of this discrepancy? This issue needs to be discussed in more detail.

5) Ultrastructural analysis of aged cells. Authors state that cells were "grown to an average age of 13" before processing for CLEM. How the age of cells was determined or confirmed?

[Editors’ note: further revisions were suggested prior to acceptance, as described below.]

Thank you for sending your article entitled "A physicochemical perspective of aging from single-cell analysis of pH, macromolecular and organellar crowding in yeast" for peer review at *eLife*. Your article is being evaluated by three peer reviewers, one of whom is a member of our Board of Reviewing Editors, and the evaluation is being overseen by Jessica Tyler as the Senior Editor.

Reviewer #1:

In this revised manuscript, the authors have addressed numerous concerns raised by reviewers, most notably, the lack of cell numbers in the analysis and the obscure title. The authors also noted additional technical difficulties for addressing all the concerns experimentally but have made appropriate revisions in the text related to these concerns. I feel that the manuscript is in an appropriate state for publication by *eLife*.

Reviewer #2:

I understand that is a revised version of the manuscript that I evaluated in February 2020. The manuscript was rejected based on the consensus of all three reviewers and it seems that the authors made an appeal to submit a revised version.

Please find below my new detailed assessment of the new manuscript.

This revised version features an extensive rewriting of the text as well as an additional experiment that is reported in Figure 1E.

I thank the reviewer for taking our comments into account by rewording extensive parts of the manuscript. Specifically, the discussion about how their data fit in with the existing literature is better than in the original version.

However, not surprisingly, the rewriting somehow tones down the significance of the results. It also highlights further the necessity to perform additional experiments, as was requested in the original review. We regret that the major concerns have not been addressed experimentally, as requested. Instead, the authors have argued that these concerns were irrelevant or fell outside of the scope of the paper. Therefore, we still have major reservations regarding the insights of this study, knowing that overall these results lead to weak conclusions and/or demonstrate that pH and macromolecular crowding are (1) likely to not to play any causative in the aging process, (2) do not appear as strong hallmarks of physiological impairments in aging, unlike other well described biological processes, and (3) require further experiments (see below) to validate the existing data sets.

The only notable exception concerns Figure 1, in which the addition of longitudinal single cell analysis with higher sampling rate provides some interesting complementary results that refine previous measurements. I think that expanding this kind of analysis, which truly exploits the capabilities of this single-cell approach would be the way to go to make this study more convincing, provided that pH/crowding are indeed demonstrated to play a role in aging. But this would require extensive additional analyses.

Last, the Introduction now suffers quite a bit from sporadic edits added during the revision, and that makes it difficult to read by now, due to number of redundancies (see specific point below).

Specific points:

1) pH measurements

– In Figure 1C, cells are grouped according to their replicative lifespans. Why is such binning achieved, since this is not exploited further (e.g. representation of the average pH drop in each group as a histogram)? Does it make sense to discriminate between 10-15 and 15-20 for instance, knowing that cells don't seem to display any different behavior?

Knowing that the average RLS of WT cells in the literature is around 25 divisions, it is surprising that there is only one bin for cells that do more than 25 divisions. Related to this, no RLS curve is displayed as a control (e.g. as supplementary figure) that shows the distribution of lifespans to prove that cells are doing well in the microfluidic device. The authors claim that cells are undergoing phototoxic damages in the experiment in Figure 1E, therefore we need to know how physiological these experiments are. This problem encountered by the authors is surprising, knowing that many other groups have reported lifespans assays using microfluidic devices that are comparable to the well-established microdissection assays.

– “We observed a gradual decrease in pH already early in life in almost all cells, and interestingly in a fraction of the cells this gradual decrease is followed by a substantial drop in pH in the subpopulation of cells that stop dividing and enter senescence (Figure 1D).”

Since the plot shows pH as a function of the number of divisions, it is impossible to assess whether pH drops substantially after cells stop dividing, unless this assessment corresponds to the very last point on each single cell trace. If so, this should be properly quantified, because it seems to only be apparent in a minority of the cells. Unlike other observations in this figure, this one is not quantified.

– “We conclude that, apart from this subpopulation of young cells with low starting pH, a shared phenotype of all aging yeast cells is that the cytosolic pH drops gradually and modestly throughout the mitotic lifespan, and that *when cells stop dividing but remain alive, the pH decreases steeply.*” (Emphasis added.)

This is neither correctly reported (see comment above) nor quantified (by averaging over a number of cells). As is, this result looks anecdotal, yet this is the main conclusion of the single cell longitudinal approach (the other result, namely, that pH drops modestly, is a recap of Kniess and Mayer, 2016).

– Figure 1—figure supplement 1C: Y label is incorrect “Fold change at age 15+/-2”. pH fold change?

If we assume that there is indeed a linear model that links pH fold change to RLS, the fit indicates that there is 5% change in pH to be expected between short lived (~10 div) and long lived (~30 div) cells. This suggests that pH is not a parameter here (hence a low correlation coefficient).

– Figure 1E: “Cytosolic acidity” is not defined. The legend says: "Normalized ratiometric pHluorin read-out (F475/F390, top) and replicative age (bottom) as a function of time for a cell that enters senescence at 26 hours (dashed blue line) after which a sharp acidification of the cytosol (dashed red line) follows."

It is normalized pH? If so, then it should be displayed as pH fold change or absolute pH values (in which case the curve would go down and not up).

Alternatively, does a doubling in acidity correspond to a 0.3 pH units decrease (as one would expect with a log10 scale)? This readout should be consistent with other pH measurements.

– Overall, it seems that pH is drop is posterior to the entry into a post-mitotic state. If true (even if partially), drawing such conclusion would be more specific than: “there is a direct link between the time spent in a slow-dividing or post-mitotic state and”, because it would imply some causality or at least a temporal ordering of events. Such temporal order could not only be done with the 36% of the cells that do arrest their cell cycle , but also with those that only slow down (most cells seem to slow down in the 3 subpopulations of cells displayed on Figure 1—figure supplement 3).

Figure 1E displays a quite sharp increase in acidity that starts after the cell cycle arrest, whereas 1F show a progressive decrease in pH. Therefore the conclusion from Figure 1 is confusing: is the evolution of pH progressive or sharp during the lifespan? As is, it is impossible to have an opinion. Since the merit of this paper it to do a longitudinal analysis, it would necessary to use a data analysis framework that clarifies this apparent discrepancy.

– Results paragraph six: 2 questions are raised but not sorted out in the Results section. Those points should be raised in the Discussion if they are not experimentally addressed in the Results section.

– The fact that pH drops in aging cells does not mean that pH homeostasis is abolished. By definition, pH homeostasis is abolished if cytosolic pH is and medium pH are equal. Here, the set point of the homeostatic system may be changed and pH homeostasis still be functional. This may sound a bit semantic, but there are many situations in biology in which it is the set point that is affected, not the homeostatic system itself, and the experiments in this manuscript do not allow to distinguish between the two. Therefore, the statement should considerably toned down, especially knowing that pH only drops by 0.5 units.

2) "Crowding" measurements:

"Plotting single-cell trajectories for cells that reach a replicative lifespan of 10, 10-15, 15-20, 20-25, or larger than 25 shows that the shortest-lived cells tend to increase the crowding levels during their lifespan, while the longer-lived cells tend to have more stable crowding levels (Figure 3C)."

This statement is far from obvious when looking at single cell data. Figure 3D and 3E show that, even if statistically significant, this effect is tiny and probably irrelevant biologically: for instance, it would be impossible to predict cell replicative age or cell lifespan based on a given N_FRET_ measurement. Therefore, I think this analysis is distracting or even useless.

"It seems that it is the maintenance of crowding homeostasis, rather than the absolute crowding levels, which has an association with lifespan"

As currently measured, the magnitude of crowding only varies by a few % throughout the lifespan. I think it is misleading to give the impression that crowding homeostasis may be impaired at all during aging. There are so many other biological processes that have reported to be massively impaired and directly connected to aging in yeast that this analysis sounds very anecdotal in comparison. A clear statement that crowding is not affected during aging would be much more clear-cut and fairer to the actual data.

In the manuscript, the authors report a change in pH from 7.5 and 7 throughout the lifespan (Figure 1). In Munder et al., 2016, decreasing the pH down to 6 or below is required to induce a transition to a gel-like cytoplasmic state. It is therefore not surprising that the authors do not observe any change in crowding.

Also, it would have been relevant to use a complementary marker of crowding , similar to what other people have used in this field (e.g. mobility measurements using microNS-GFP, Gln1-GFP, etc.) , in order to check the results obtained with the FRET sensor, especially knowing that the obtained results are mostly negative. In particular, these complementary measurements would somehow make the link between the macromolecular crowding and organelle crowding which lead to opposing conclusions in the present study.

By the way, “crowding” somehow refers to a functional property: its measurement allows one to assess whether diffusive and transport processes are impaired. Here, the authors use a slightly different meaning, for instance by measuring distance between organelles. One may question whether a decrease in organelles distance necessarily leads to functional impairments. This would need to be assessed using mobility assays (as mentioned above).

3) Introduction issues/redundancies:

The Introduction lacks structure: background information are mixed with justification of the choices made to study pH/crowding in the study. There are several redundancies in the Introduction that make it confusing and superficial, because the same arguments/information are repeated over and over.

– “we select crowding and pH”

Macromolecular crowding should be precisely defined and justified with relevant references, especially if the paper is intended to be read in part by aging experts.

– Introduction paragraph three: using both “alkalization” and “acidification” makes this paragraph quite confusing. Specifically, since this paragraph somehow points out the controversy about the evolution of pH during replicative aging, instead of: “the pH needs to be considered when defining a physicochemical roadmap of cellular aging” , I would rather write that measurements of pH during aging lead to opposite conclusions, therefore, additional data are required to sort out the controversy.

“what is currently missing is a single cell perspective on cytosolic pH in yeast replicative ageing”: this statement still lacks justification: why is it essential to perform complementary measurements? This is not explained… Instead, the authors the authors use the word “physicochemical roadmap” which does not explain the underlying controversy.

These 3 sentences are redundant:

“both properties, in addition to affecting the function of individual key molecules, also interplay to impact intracellular organization”

“pH have the potential to drive profound changes to intracellular organization (reference to Munder et al., 2016)”

“The pH also influences macromolecular organization in yeast:” (reference to Munder et al., 2016 again…)

4) Discussion

I think the additional discussion is fair, i.e. the fact that the modest cytosol acidification is likely to be a consequence of upstream impaired biological processes.

“We show that cytosolic acidity strongly increases only after entry into senescence and we do not observe drastic changes in early lifespan.” I'm ok with this, yet Figure 1F still yields an opposite conclusion. This would have to be clarified further (see point 1) above).

5) Comments on the rebuttal letter

In the Highlights section:

"the finding that crowding on the scale of macromolecules is well maintained in ageing implying that is must be strictly regulated”

I'm afraid that this is a purely rhetorical argument, that does not provide a very strong argument in favor of the paper. The fact that X or Y is not impaired during aging does unfortunately not help understand the mechanisms driving aging.

"the first single cell time course measurements of cytosolic pH in ageing revealing that the timing and direction of pH changes in cytosol, vacuoles, and cortical pH are distinct"

This is misleading because the paper does not properly address this question: only cytosolic measurements are performed in this study, thus a direct comparison to vacuolar and cortical pH is impossible based on the proposed datasets. It is only through a comparison to previous literature, which makes this conclusion much less strong.

Therefore, the controversy about vacuolar (apparently an early-life pH increase) versus cytosolic (presumably a late-life pH decrease) pH dynamics cannot be addressed based on the data reported in this manuscript. This means that point #2 from reviewer #1, point #3 from reviewer #3, and point #4 from reviewer #3 are left unaddressed.

Regarding the link between either pH or crowding and aging, the magnitude of effects observed in young versus old cells is very small and tend to support a model in which these physicochemical parameters are either not involved in the entry into senescence, or enslaved to upstream biological processes that truly affect cellular fitness (monitored as cell cycle arrest/slow down). Therefore, knowing that there is not technical or conceptual breakthrough in this paper, one may really question whether it adds any relevant insight towards a better understanding of the aging process.

"Organellar crowding is strongly increased. We highlighted the impact on diffusion times of cellular structures such as ribosomes and RNP granules, and the possible increase in membrane contact sites. These are important findings"

The argument about diffusion times is purely theoretical and would require an experimental validation (cf. comment about microNS-GFP above) to make the point.

Reviewer #3:

All my concerns have been addressed by the authors in the revised manuscript. I support publication in its current form.

---

## [Author Response]

[Editors’ note: The authors appealed the original decision. What follows is the authors’ response to the first round of review.]

Thank you for reviewing our work now entitled "A physicochemical perspective of aging from singlecell analysis of pH, macromolecular and organellar crowding in yeast” and the invitation to submit our manuscript for revision. We have carefully considered the valuable comments by the reviewers. The possibilities to perform experiments are currently limited due to the global pandemic, but we have managed to perform all experiments that contribute to the thoroughness of the publication, and have, where appropriate, adjusted the text to account for the remaining concerns. We hope you agree that the manuscript in its current form is suitable for the special edition on aging in *eLife*. The highlights are:

– a valuable high resolution EM dataset showing for the first time that organellar crowding increases dramatically in ageing,

– the finding that crowding on the scale of macromolecules is well maintained in ageing implying that is must be strictly regulated,

– a newly developed crowding sensor that will aid to dissect the above mechanism in future studies,

– the first single cell time course measurements of cytosolic pH in ageing revealing that the timing and direction of pH changes in cytosol, vacuoles, and cortical pH are distinct and pointing out a possible role in pH homeostasis for the proteome.

Reviewer #1:In the manuscript titled "A physicochemical roadmap of yeast replicative aging", authors Mouton et al. measured cytosolic pH, cellular crowding levels, and organellar volume in replicatively aged yeast cells, compared to young ones. Using a novel cellular pH sensor, the authors reported reduced pH in the cytosol in aged cells. Using a set of FRET-based crowding probes, the authors found no apparent change in cellular macromolecular crowding during aging. Using correlative light and electron microscopy, the authors measured the volume of vacuole and nuclei. This analysis showed both increased volume for vacuole and nuclei and increased inter-organellar crowding.Overall, the authors used novel technologies and measured a number of cellular physiochemical changes during yeast replicative aging, providing a set of experimental data that are not previously available. However, there are a few key weaknesses even when considering this manuscript as a resources paper without investigating in the mechanistic details.1) Several assays only measured a very limited number of cells, especially for the cytosolic pH, where only 30 cells were analyzed. Importantly, as the authors noted, around 50% of these cells were able to maintain a pH in the normal range of young cells. This percentage appears to be even higher for cells that have relatively longer replicative lifespan. Given the broad variation in this phenotype, more cells should be included in this analysis.

We agree that the numbers of cells analyzed were somewhat on the low side and have increased the number to 80 for the single cell pH and the crowding measurements (adjusted Figures 1C,D,F and 3B-E and Figure 1—figure supplement 1 and Figure 3—figure supplement 1). Increasing the dataset has not changed the numbers such as population averages, means, the statistics and correlations nor the conclusions drawn.

2) Vacuole acidity is much reduced in aged cells due to imbalance in proton pumps between vacuolar membranes and plasma membranes (Hughes and Gottschling, 2012). Could this imbalance be the cause of cytosol acidification? Does cytosol acidification and vacuole deacidification concur at the single-cell level during aging? VMA1 overexpression has been shown to at least partially correct the proton pump imbalance resulting in extended lifespan. Does it also ameliorate the cytosol acidification during aging? Many of these questions can be experimentally tested.

We agree and we have now addressed these issues better. We included new data to elucidate the precise timing of the cytosolic changes (Figure 1E and Figure 1—figure supplement 2 and 3 and text in subsection “Yeast replicative aging leads to acidification of the cytosol, especially after entry into senescence”) and we discuss on how this related to those in the vacuole and cell cortex (subsection “pH homeostasis in aging”). Whether one should expect that VMA1 overexpression would be sufficient to ameliorate the cytosol acidification is not so obvious to us considering that the timing of pH changes is rather different in cytosol and vacuole. While it would indeed be an interesting experiment, a practical concern is that our microfluidic chips are designed for haploid cells while the overexpression of VMA1 should be done in diploids as the haploids are sick and grow very slow.

3) CLEM analysis is another assay that was only performed for a very small number of cells (14 aged cells and 10 young cells). Although hundreds of section images were analyzed, they still only represent a very few selected cells. Given the general huge variation in cellular morphology, these small numbers are not sufficient to make a convincing case.

We respectfully disagree that the presented data are insufficient to make our point, which is, that organellar crowding increases dramatically in aged cells. This is a new finding that can only be obtained from the kind of (low throughput) EM data presented here. The cells chosen for analysis were selected randomly and the effects are statistically significant and unambiguous. While we agree that there is a high degree of cell-to-cell variation for some age-associated phenotypes which requires larger cohort sizes, for this particular phenotype the effect size is sufficiently large and the variation is sufficiently small that it is easily detectable with the number of cells analyzed here.

The CLEM analysis of aged cells is very challenging and not high throughput as aged cells make up only a minor fraction of the exponentially growing population. At the 20 hours time point used here, each aged (and labeled) cell is outnumbered by thousands of daughter cells. In the current experimental set, we chose to age the yeast cells in normal exponential growing cultures. This setup has minimal interventions (no attachments of beads for enriching aged cell, no genetic programs to enrich for aged cells, etc.) and hence we can accurately and uniquely report the ultrastructure of ageing cells.

4) For the CLEM experiment, the authors mentioned that the cells were aged "over 20 hours to achieve an estimated replicative age of 13 divisions". Since these cells were imaged by fluorescent microscopy, would it be possible to precisely determine their replicative age by staining and counting bud scars? It seems to make sense to count bud scars for these 14 old cells and 10 young cells.

This is unfortunately not possible. In our previous work we addressed the spread in the ages of cells cultured for 20 hours: we had modelled (by considering the distribution of division times) and experimentally determined (performing bud scar counts) the age-distributions and find that the ages in the population are approximately normally distributed with a half maximum peak width of 10 divisions (Janssens et al., 2015, Figure 1—figure supplement 2). These studies have shown us that we can estimate the age distribution at 20 hours based on the doubling time of the cells. The current studies were performed in rich medium where the doubling times are faster, hence the average of 13. Also, in the microfluidic setup we observe a similar distributing of ages after 20 hours.

In the current text we have made the uncertainty of the exact ages of the represented cells explicit in the Results section.

5) The authors stated that "14 tomograms were acquired and segmented to define the plasma membrane, nucleus, vacuole, lipid droplets, ER, multivesicular bodies, and mitochondria". Since these fine cellular structures, other than vacuole and nucleus, are clearly defined, why didn't the authors measure and compare them? Age-associated morphological changes to plasma membrane, lipid droplets, ER, vesicles, and mitochondria at the EM resolution are likely provide more insights to the physiology state of aging cells and support their conclusions based on the analysis of vacuole and nucleus.

The importance of simultaneous visualization of all organellar membranes enabled the quantification of the distances between organelles, and hence the organellar crowding. Quantification of individual features like surface area or volume of the ER, Mitochondria and MVB’s is not possible considering that we could quantify only few of each type. We did include an analysis of the number, and surface area of lipid droplets, showing lipid droplets increase in ageing (Figure 4—figure supplement 3).

Reviewer #2:In this study, Mouton et al. use longitudinal tracking of single dividing yeast cell to followthe dynamics of several physiological markers using fluorescence microscopy in aging cells. Using the ratiometric probe phluorin, they report a gradual loss of pH in aging cells, that is not predictive of cell death. Then, they use a modified version of a molecular crowding sensor and report that crowding homeostasis is maintained during aging. Last, they perform cellular EM tomographic analyses to monitor the evolution of organelles size with age and they show that inter-organelles crowding increases with age.Major concerns:– The message conveyed by the paper is somewhat blurry, knowing that pH measurements do not appear to be conclusive: the authors confirm previous results showing a decrease in pH with age, but they show that it does not predict the evolution of cell fate. Also, they do not provide evidence that this observation is connected to any other well-described physiological marks of age (e.g. protein aggregates, ERCs, etc.). Similarly, they show that molecular crowding does not seem to be a marker of replicative aging. Therefore, even though some observations are novel in the context of aging (measurements of molecular and organelles crowding), the paper is lacking insights of high biological significance regarding the mechanism that controls the entry into replicative senescence.

We are sorry to read we have not conveyed the novelty of our paper well enough. We have made several textual changes to explain this better in the current manuscript, namely:

i) The single-cell data showing the relationship between cytosolic pH and replicative age is new and not merely confirmatory to previous data which were based on population-level data. It is this single-cell perspective that reveals several aspects that remain invisible on the population levels (the timing of changes, the variation within the population, correlations with lifespan). The novelty and implications of our cytosolic pH data is now better discussed in subsection “pH homeostasis in aging”.

ii) Macromolecular crowding is well maintained throughout ageing and future studies should be aimed at finding what regulates this homeostasis. This is completely uncharted territory and outside the scope of this paper. The newly developed crowding sensor will provide a valuable tool identifying the genes that regulate crowding

iii) Organellar crowding is strongly increased. We highlighted the impact on diffusion times of cellular structures such as ribosomes and RNP granules, and the possible increase in membrane contact sites. These are important findings, which imply that out of the available methods to study crowding, which are tracer particles and crowding sensors, results obtained through tracer particles have to be carefully interpreted, because of confinement artifacts. These observations also warrant future investigations that fall outside the scope of this paper.

We respectfully disagree that additional evidence is needed showing that homeostasis of pH and crowding are connected to other well-described physiological marks of age (e.g. protein aggregates); we reference many papers that establish how volume regulation, pH homeostasis and crowding impact biology. We consider establishing how they are connected beyond the scope of this paper.

– One weakness of the paper is that it appears as an assembly of disparate observations that are wrapped as a “physicochemical roadmap”. This term sounds quite obscure, and a bit oversold, or sophistic, knowing that it is only supported by a limited set of readouts that provide mostly inconclusive results.

Our revised manuscript highlights the many clear connections between the separate observations throughout the text; the observations are intimately related with each other. However, we do not want our title to sound obscure or come across as oversold and thus have adjusted it to the more precise “A physicochemical perspective of aging from single-cell analysis of pH, macromolecular and organellar crowding in yeast”.

– pH measurements in aging cells have already performed, leading to the same conclusion that pH decreases with age (using population-based measurements, Kniess et al., 2016, as acknowledged by the authors). However, another study (Henderson et al., 2014, also cited in the present manuscript) also reported measurements of “cortex-proximal cytosolic pH” in aging cells. In that paper, an increase in pH was observed. Whether the cortex-proximal cytosol has an identical pH as the “overall” cytosol is unclear at this point.Therefore, knowing that the main merit of the paper is to attempt to confirm previous measurements of cytosolic pH, one would expect the authors to clearly address this controversy using in-depth complementary experiments. Instead, interpretations of results are made in such a way to give the impression that the results obtained by the authors is in line with all the literature, which is misleading.

We realize we might have left a wrong impression for the merits of our manuscript. The aim of our work was to show the different physicochemical environment that aged cytosol of yeast cells have compared to young cells (now stated in the Introduction). We did not aim to resolve long standing questions in the field regarding pH of different compartments and we apologize if we did not express our intentions better. In order to reliably measure crowding in ageing cells we needed single-cell data on the evolution of the cytosolic pH, which was not yet available.

We have now adjusted our Discussion to address better how our findings relate to previously published pH measurements at the cell cortex. Please see our answers to reviewer 1 point 2.

In this context, this statement in the Discussion: “Here, we provide the first (to our knowledge) roadmap for the progressive decline in aging in several parameters defining a cell's intracellular environment, namely, pH, crowding, and volume, all of which impinge on the hallmarks of aging” is inappropriate, knowing that both pH and volume have already been documented in aging cells.

We have removed the sentence and replaced it with the more precise “Here, we provide an analysis of the progressive change during aging for several parameters that define a cell’s intracellular environment; namely, cytosolic pH and crowding on the scale of macromolecules and organelles, all impinging on the hallmarks of aging.”

Reviewer #3:In this manuscript, Mouton et al. have used a combination of single cell imaging and electron microscopy to survey heterogeneity in age-dependent physiological changes, including cytosolic pH, macromolecular crowding, and organellar size, during replicative aging in yeast. This manuscript addresses an important and interesting topic that has not yet been thoroughly investigated by previous studies. The authors made an interesting observation that in replicatively-aged yeast cytosolic pH gradually decreases early in life, but undergoes a significant drop when the cells stop dividing. However, at the single cell level old cells demonstrate incomplete penetrance phenotype, with only ~ 50% of the old cells showing this significant pH drop. Strikingly, despite significant increase in cell volume with age and increased organellar density, which authors refer to as "organellar crowding", macromolecular crowding remained very stable through the cell lifespan. This study is rigorously designed and of good technical quality. Although the exact mechanism for the incomplete penetrance remains unclear, this paper reports an initial framework and important technical resources/methodology that could significantly accelerate studies addressing heterogeneity of the aging process. I would support publication of the manuscript in eLife as Tools and Resources. However, several points need to be further clarified by the authors.1) One of the concerns is the low number of cells that were used for analysis of the pH and FRET reporters using microfluidics. Several hundreds of cells can be typically trapped and monitored though their replicative lifespan in the microfluidic device. Were any of the cells excluded from the analysis? If yes, what were the criteria for exclusion? An increase in the number of cells analyzed would help to better assess the proportion of cells that experience a drop in pH and strengthen their conclusions.

Indeed, many cells can be trapped in the device, although the absolute numbers vary per experiment. However, the number of cells that remain in the device from the start until their death is significantly lower compared to the total number of trapped cells. At each imaging position we have included all cells that are imaged from start to death and have not excluded cells from the analysis.

As mentioned above, we have increased the numbers of cells analyzed for both cytosolic pH and macromolecular crowding to 80 cells. Further increase in these numbers seems unnecessary, since none of our conclusions or their significance have changed after the additional analysis.

2) Another limitation of this study is that time intervals selected to collect fluorescent images for aging experiments (every 10 hours) do not allow assessing short-term physiological fluctuations. For example, does pH change in individual cells between different phases of the cell cycle? This issue needs at least to be discussed.

Imaging at higher time density runs the risk of increased phototoxicity and as our aim was to follow age-related phenotypes in pH, we chose to not measure short-term fluctuations in physiology. We have now generated a new dataset in which we follow cytosolic pH every hour in order to address the timing of pH drop better (see Figure 1E and Figure 1—figure supplement 2 and 3 and Supplementary file 3 confirming the mild impact of higher frequency imaging on lifespan). We agree that the issue of pH changes in the cell cycle should have been discussed and we now do so in Results paragraph four. The new data shows that variations in pH as function of the cell cycle are minor compared to those in aging.

3) The Introduction section could be improved by logically putting the findings of this paper in the context of the published studies. Does the cell size (organellar volume) affect intracellular (organellar pH)? In several places, authors' statements are not supported by proper citations. E.g.: "In healthy young yeast cells, a "volume hierarchy" is observed where the cytosol represents the largest compartment, followed by the nucleus and vacuole." And "V-ATPase pumps protons from the cytosol into the lumen of various organelles and regulates their pH".

Thank you for pointing this out. We have added the references and clarified connections to the existing literature in the revised Introduction and Discussion.

4) Another issue is related to the interpretation of the results. Although the data presented in this manuscript is not sufficient to assess pH changes in different cellular compartments (only in cytosol), authors should discuss whether the loss of vacuolar acidity with aging (Hughes and Gottschling, 2012) can potentially precede and/or be causative for the observed drop in cytosolic pH in old cells. Moreover, the decrease of cytosolic pH with aging (Knieß and Mayer, 2016) seems to contradict with previously reported data showing increased pH in the cell cortex of old mother cells (Henderson et al., 2014). What is the author interpretation of this discrepancy? This issue needs to be discussed in more detail.

We have indeed added a discussion related to these previous findings. Please also refer to our answer to reviewer 1 point 2.

5) Ultrastructural analysis of aged cells. Authors state that cells were "grown to an average age of 13" before processing for CLEM. How the age of cells was determined or confirmed?

Please see our answer to reviewer 1 point 4.

[Editors’ note: what follows is the authors’ response to the second round of review.]

Reviewer #2:[…]Specific points:1) pH measurements– In Figure 1C, cells are grouped according to their replicative lifespans. Why is such binning achieved, since this is not exploited further (e.g. representation of the average pH drop in each group as a histogram)? Does it make sense to discriminate between 10-15 and 15-20 for instance, knowing that cells don't seem to display any different behaviour ?Knowing that the average RLS of WT cells in the literature is around 25 divisions, it is surprising that there is only one bin for cells that do more than 25 divisions. Related to this, no RLS curve is displayed as a control (e.g. as supplementary figure) that shows the distribution of lifespans to prove that cells are doing well in the microfluidic device. The authors claim that cells are undergoing phototoxic damages in the experiment in Figure 1E, therefore we need to know how physiological these experiments are. This problem encountered by the authors is surprising, knowing that many other groups have reported lifespans assays using microfluidic devices that are comparable to the well-established microdissection assays.– “We observed a gradual decrease in pH already early in life in almost all cells, and interestingly in a fraction of the cells this gradual decrease is followed by a substantial drop in pH in the subpopulation of cells that stop dividing and enter senescence (Figure 1D).”Since the plot show pH as a function of the number of divisions, it is impossible to assess whether pH drops substantially after cells stop dividing, unless this assessment corresponds to the very last point on each single cell trace. If so, this should be properly quantified, because it seems to only be apparent in a minority of the cells. Unlike other observations in this figure, this one is not quantified.– ““We conclude that, apart from this subpopulation of young cells with low starting pH, a shared phenotype of all aging yeast cells is that the cytosolic pH drops gradually and modestly throughout the mitotic lifespan, and that when cells stop dividing but remain alive, the pH decreases steeply.” (Emphasis added)This is neither correctly reported (see comment above) nor quantified (by averaging over a number of cells). As is, this result looks anecdotal, yet this is the main conclusion of the single cell longitudinal approach (the other result, namely, that pH drops modestly, is a recap of Kniess and Mayer, 2016).– Figure 1—figure supplement 1C: Y label is incorrect “Fold change at age 15+/-2”. pH fold change?If we assume that there is indeed a linear model that links pH fold change to RLS, the fit indicates that there is 5% change in pH to be expected between short lived (~10 divv) and long lived (~30 div) cells. This suggests that pH is not a parameter here (hence a low correlation coefficient).– Figure 1E: “Cytosolic acidity” is not defined. The legend says: "Normalized ratiometric pHluorin read-out (F475/F390, top) and replicative age (bottom) as a function of time for a cell that enters senescence at 26 hours (dashed blue line) after which a sharp acidification of the cytosol (dashed red line) follows."It is normalized pH? If so, then it should be displayed as pH fold change or absolute pH values (in which case the curve would go down and not up).Alternatively, does a doubling in acidity correspond to a 0.3 pH units decrease (as one would expect with a log10 scale)? This readout should be consistent with other pH measurements.– Overall, it seems that pH is drop is posterior to the entry into a post-mitotic state. If true (even if partially), drawing such conclusion would be more specific than: “there is a direct link between the time spent in a slow-dividing or post-mitotic state and”, because it would imply some causality or at least a temporal ordering of events. Such temporal order could not only be done with the 36% of the cells that do arrest their cell cycle , but also with those that only slow down (most cells seem to slow down in the 3 subpopulations of cells displayed on Figure 1—figure supplement 3).Figure 1E displays a quite sharp increase in acidity that starts after the cell cycle arrest, whereas 1F show a progressive decrease in pH. Therefore the conclusion from Figure 1 is confusing: is the evolution of pH progressive or sharp during the lifespan? As is, it is impossible to have an opinion. Since the merit of this paper it to do a longitudinal analysis, it would necessary to use a data analysis framework that clarifies this apparent discrepancy.– Results paragraph six: 2 questions are raised but not sorted out in the Results section. Those points should be raised in the Discussion if they are not experimentally addressed in the Results section.– The fact that pH drops in aging cells does not mean that pH homeostasis is abolished. By definition, pH homeostasis is abolished if cytosolic pH is and medium pH are equal. Here, the set point of the homeostatic system may be changed and pH homeostasis still be functional. This may sound a bit semantic, but there are many situations in biology in which it is the set point that is affected, not the homeostatic system itself, and the experiments in this manuscript do not allow to distinguish between the two. Therefore, the statement should considerably toned down, especially knowing that pH only drops by 0.5 units.2) "Crowding" measurements:"Plotting single-cell trajectories for cells that reach a replicative lifespan of 10, 10-15, 15-20, 20-25, or larger than 25 shows that the shortest-lived cells tend to increase the crowding levels during their lifespan, while the longer-lived cells tend to have more stable crowding levels (Figure 3C)."This statement is far from obvious when looking at single cell data. Figure 3D and 3E show that, even if statistically significant, this effect is tiny and probably irrelevant biologically: for instance, it would be impossible to predict cell replicative age or cell lifespan based on a given N_FRET_ measurement. Therefore, I think this analysis is distracting or even useless."It seems that it is the maintenance of crowding homeostasis, rather than the absolute crowding levels, which has an association with lifespan"As currently measured, the magnitude of crowding only varies by a few % throughout the lifespan. I think it is misleading to give the impression that crowding homeostasis may be impaired at all during aging. There are so many other biological processes that have reported to be massively impaired and directly connected to aging in yeast that this analysis sounds very anecdotal in comparison. A clear statement that crowding is not affected during aging would be much more clear-cut and fairer to the actual data.In the manuscript, the authors report a change in pH from 7.5 and 7 throughout the lifespan (Figure 1). In Munder et al., 2016, decreasing the pH down to 6 or below is required to induce a transition to a gel-like cytoplasmic state. It is therefore not surprising that the authors do not observe any change in crowding.Also, it would have been relevant to use a complementary marker of crowding , similar to what other people have used in this field (e.g. mobility measurements using microNS-GFP, Gln1-GFP, etc.) , in order to check the results obtained with the FRET sensor, especially knowing that the obtained results are mostly negative. In particular, these complementary measurements would somehow make the link between the macromolecular crowding and organelle crowding which lead to opposing conclusions in the present study.By the way, “crowding” somehow refers to a functional property: its measurement allows one to assess whether diffusive and transport processes are impaired. Here, the authors use a slightly different meaning, for instance by measuring distance between organelles. One may question whether a decrease in organelles distance necessarily leads to functional impairments. This would need to be assessed using mobility assays (as mentioned above).3) Introduction issues/redundancies:The Introduction lacks structure: background information are mixed with justification of the choices made to study pH/crowding in the study. There are several redundancies in the Introduction that make it confusing and superficial, because the same arguments/information are repeated over and over.– “we select crowding and pH”Macromolecular crowding should be precisely defined and justified with relevant references, especially if the paper is intended to be read in part by aging experts– Introduction paragraph three: using both “alkalization” and “acidification” makes this paragraph quite confusing. Specifically, since this paragraph somehow points out the controversy about the evolution of pH during replicative aging, instead of: “the pH needs to be considered when defining a physicochemical roadmap of cellular aging”, I would rather write that measurements of pH during aging lead to opposite conclusions, therefore, additional data are required to sort out the controversy.“what is currently missing is a single cell perspective on cytosolic pH in yeast replicative ageing”: this statement still lacks justification: why is it essential to perform complementary measurements ? This is not explained… Instead, the authors the authors use the word “physicochemical roadmap” which does not explain the underlying controversy.These 3 sentences are redundant:“both properties, in addition to affecting the function of individual key molecules, also interplay to impact intracellular organization”“pH have the potential to drive profound changes to intracellular organization (reference to Munder et al., 2016)”“The pH also influences macromolecular organization in yeast:” (reference to Munder et al., 2016 again…)4) DiscussionI think the additional discussion is fair, i.e. the fact that the modest cytosol acidification is likely to be a consequence of upstream impaired biological processes.“We show that cytosolic acidity strongly increases only after entry into senescence and we do not observe drastic changes in early lifespan.” I'm ok with this, yet Figure 1F still yields an opposite conclusion. This would have to be clarified further (see point 1) above).5) Comments on the rebuttal letterIn the Highlights section:"the finding that crowding on the scale of macromolecules is well maintained in ageing implying that is must be strictly regulated”I'm afraid that this is a purely rhetorical argument, that does not provide a very strong argument in favor of the paper. The fact that X or Y is not impaired during aging does unfortunately not help understand the mechanisms driving aging. IN addition, there are likely many processes that"the first single cell time course measurements of cytosolic pH in ageing revealing that the timing and direction of pH changes in cytosol, vacuoles, and cortical pH are distinct"This is misleading because the paper does not properly address this question: only cytosolic measurements are performed in this study, thus a direct comparison to vacuolar and cortical pH is impossible based on the proposed datasets. It is only through a comparison to previous literature, which makes this conclusion much less strong.Therefore, the controversy about vacuolar (apparently an early-life pH increase) versus cytosolic (presumably a late-life pH decrease) pH dynamics cannot be addressed based on the data reported in this manuscript. This means that point #2 from reviewer #1, point #3 from reviewer #3, and point #4 from reviewer #3 are left unaddressed.Regarding the link between either pH or crowding and aging, the magnitude of effects observed in young versus old cells is very small and tend to support a model in which these physicochemical parameters are either not involved in the entry into senescence, or enslaved to upstream biological processes that truly affect cellular fitness (monitored as cell cycle arrest/slow down). Therefore, knowing that there is not technical or conceptual breakthrough in this paper, one may really question whether it adds any relevant insight towards a better understanding of the aging process."Organellar crowding is strongly increased. We highlighted the impact on diffusion times of cellular structures such as ribosomes and RNP granules, and the possible increase in membrane contact sites. These are important findings"The argument about diffusion times is purely theoretical and would require an experimental validation (cf. comment about microNS-GFP above) to make the point.

We are pleased to read that reviewers #1 and #3 have no further concerns and support publication of the manuscript in its current state. After careful reading of the review of reviewer #2 we have made the following changes.

1) As we interpret it, the reviewer suggests in the main text that we obtain even more single-cell data at a higher time resolution (alike the new data in Figure 1E). This is not interesting to do for the measurements of macromolecular crowding as the levels are relatively stable in aging, and, it is not feasible to do so for the organellar measurements as previously explained: CLEM is unsuited for this.

2) Related to Figure 1. The text suggestions have been incorporated. Other comments are explained below:

1) Why is data presented in age groups?

To avoid overcrowded panels, we distribute the data over multiple panels; grouped to RLS is the most logical way.

2) Figure 1E plots as "acidity instead of pH"

This is because the vacuolar data from (Chen et al., 2020) had not been calibrated to pH but only reports the direct read-out of the pH sensor.

3) What is RLS in this chip design?

There is no difference between the replicative lifespan determined in this microfluidic device or by microdissection; the reviewer can find this specific information in Figure 2E, published by (Crane et al., 2014), also confirmed in our hands in (Janssens, et al., 2016; Rempel et al., 2019).

4) It is confusing that Figure 1E displays a sharp increase in acidity, whereas 1F shows a progressive decrease in pH.

In Figure 1E, we show cytosolic acidity on the y-axis as in (Chen et al., 2020), and in Figure 1F, we show average cytosolic pH over the whole population, therefore the appearance of gradual changes.

5) Sharp decrease in pH of post-mitotic cells should be quantified “by averaging over a number of cells”.

In fact, we had already quantified the variable the reviewer is asking for (Figure 1C, specifically, and all raw data provided in Figure 1). Because the decrease in pH before senescence is only minor (as can be seen from Figure 1D), comparing the first and last measurement points of all cells (as done in 1C) is sufficient.

3) Related to Figure 3. We have included a sentence to the Abstract that crowding is rather stable in aging. Other comments are explained below:

1) Figure 3D and 3E show that, even if statistically significant, the relation between crowding and lifespan is tiny and probably irrelevant biologically.

We agree that crowding remains rather stable in aging. We have stated that numerous times throughout the manuscript, but we have added it back again to the Abstract as well. However, we respectfully disagree that the presentation of the details in Figure 3D and E are unimportant. The changes displayed in Figure 3D and E are within 4%. Considering that maximum change of crowding under severe osmotic shock is ~20%, and recovers within minutes, 4% cannot be considered minor or biologically irrelevant.

2) The reviewer suggests complementing the macromolecular crowing measurements and the organellar crowding measurements with measurements of mobility (single-particle tracking with μNS) or self-assembly of Gln1.

μNS particles are 10 times bigger than the crowding sensor, thus probing crowding on a completely different scale than reported in our study. Performing such experiments would not confirm or contradict any of our findings, since crowding effects are size-dependent (Delarue et al., 2018). Self-assembly of Gln1 could be a good complementary experiment, but also here, such experiments would not confirm or contradict any of our findings, but rather probe an additional parameter.

3) Figure 4—figure supplement 4, referred to in the Discussion requires experimental validation with μNS particles.

Given the data presented in Figure 5, the mobility of fluorescent nanoparticles particles must be decreased due to organellar crowding: Such enormous obstacles would hamper diffusion.

4) Textual issues related to the Introduction and Discussion have been addressed.